# SHAPEX : Shapelet-Driven Post Hoc Explanations for Time Series Classification Models

**Bosong Huang[1], Ming Jin[1]\*, Yuxuan Liang[2], Johan Barthelemy[3], Debo Cheng[4],**
**Qingsong Wen[5], Chenghao Liu[6], Shirui Pan[1]\***
[1]Griffith University  [2]Hong Kong University of Science and Technology (Guangzhou)  [3]NVIDIA
[4]Hainan University  [5]Squirrel Ai Learning, USA  [6]Salesforce Research Asia
bosong.huang@griffithuni.edu.au, mingjinedu@gmail.com, s.pan@griffith.edu.au

## Abstract

Explaining time series classification models is crucial, particularly in high-stakes applications such as healthcare and finance, where transparency and trust play a critical role. Although numerous time series classification methods have identified key subsequences, known as shapelets, as core features for achieving state-of-the-art performance and validating their pivotal role in classification outcomes, existing post-hoc time series explanation (PHTSE) methods primarily focus on timestep-level feature attribution. These explanation methods overlook the fundamental prior that classification outcomes are predominantly driven by key shapelets. To bridge this gap, we present SHAPEX, an innovative framework that segments time series into meaningful shapelet-driven segments and employs Shapley values to assess their saliency. At the core of SHAPEX lies the Shapelet Describe-and-Detect (SDD) framework, which effectively learns a diverse set of shapelets essential for classification. We further demonstrate that SHAPEX produces explanations which reveal causal relationships instead of just correlations, owing to the atomicity properties of shapelets. Experimental results on both synthetic and real-world datasets demonstrate that SHAPEX outperforms existing methods in identifying the most relevant subsequences, enhancing both the precision and causal fidelity of time series explanations. Our code is made available at https://github.com/BosonHwang/ShapeX

## 1 Introduction

Time series classification plays a critical role across various domains [1]. While deep learning models have significantly advanced classification performance in this area, their black-box nature often compromises explainability. This limitation becomes particularly critical in high-stakes applications such as healthcare and finance, where reliability and accuracy are paramount. Most research on time-series model explainability focuses on post-hoc time-series explanation (PHTSE), predominantly via perturbation-based methods at the timestep level. For example, Dynamask [2] generates instance-specific importance scores through perturbation masks. Similarly, TIMEX [3] trains a surrogate model to mimic pretrained model behavior, whereas TIMEX++ [4] leverages information bottleneck to deliver robust explanations.

Despite these advancements, existing methods neglect an essential aspect of time series classification: the classification outcomes are often driven by *key subsequences* rather than individual timesteps. Recently, several classification methods [5, 6, 7] have emphasized the significance of utilizing these crucial subsequences, known as *shapelets*. Take some examples for intuitive understanding: In electrocardiogram (ECG) signal classification, specific segments like the QRS com-

---

\*Corresponding authors.

plex are critical for accurate diagnostics [8]. In human activity recognition (HAR), patterns like gait cycles play a decisive role in classifying activities [9]. Recent methods have expanded upon this foundational knowledge, exploring diverse strategies to incorporate shapelets into classification frameworks. For instance, ShapeFormer [6] employs a discovery algorithm to extract class-specific shapelets and incorporates them into a transformer-based model to enhance classification performance. Wen et al. [7] generalizes shapelets into symbolic representations, enhancing flexibility and expressiveness. Similarly, Qu et al. [5] demonstrates that convolutional kernels can naturally function as shapelets, offering both strong discriminative power and interpretability. Together, these approaches underscore a growing consensus that shapelets are not only interpretable but also central to the performance of modern time series classification models.

Although shapelets are widely recognized as key features in time series classification, existing explanation methods primarily rely on perturbing individual timesteps to compute saliency scores. This disrupts local temporal dependencies and fails to capture the holistic influence of key subsequences, resulting in fragmented and less interpretable explanations. As shown in Figure 1 (b), timestep-level perturbations are scattered, making it difficult to align with true saliency patterns. While equal-length segment perturbations (Figure 1 (c)) may further reduce interpretability by splitting meaningful patterns or merging unrelated subsequences. To overcome these limitations, an ideal approach should perturb segments aligned with meaningful shapelets, as illustrated in Figure 1 (d). This preserves the atomicity of key

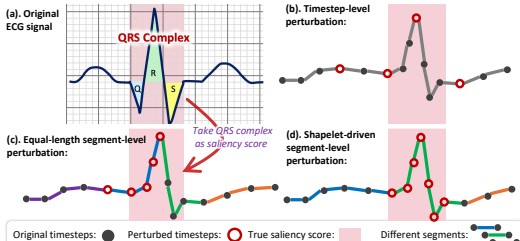

Figure 1: Perturbation strategies on an ECG example. (a) shows the original signal with saliency labels derived from QRS complexes—clinically critical regions for arrhythmia classification. (b)–(d) apply timestep-level, equal-length, and shapelet-driven perturbations, respectively. Only (d) maintains alignment with the salient regions.

subsequences, potentially leading to more faithful and interpretable explanations. However, existing time series segmentation methods primarily detect statistical change points [10, 11] or extract subsequences unrelated to determining classification outcomes [12], making them unsuitable for explaining model behaviors.

To fill the gaps, we introduce SHAPEX, a novel approach that segments the time series into meaningful subsequences and computes Shapley value [13] as saliency scores. Instead of distributing importance across individual timesteps, SHAPEX aggregates timesteps into cohesive, shapelet-driven segments that serve as "players" in the Shapley value computation. By measuring each segment's marginal contribution to the black-box model's prediction, this method clearly identifies which subsequences significantly influence classification outcomes. Specifically, the Shapelet Describe-and-Detect (SDD) framework within SHAPEX efficiently discovers representative and diverse shapelets most critical for classification. Subsequently, SHAPEX segments the time series to align closely with these dominant temporal patterns, yielding more accurate and interpretable explanations. Our key contributions are summarized as follows:

1. SHAPEX pioneers PHTSE at the shapelet-driven segment level, offering a precise and principled alternative to timestep-level methods. Its SDD framework is the first shapelet learning approach designed specifically for PHTSE, enforcing representativeness and diversity.

2. We provide a theoretical justification for interpreting the Shapley value computed by SHAPEX as an approximation of the model-level Conditional Average Treatment Effect (CATE), thereby enabling more trustworthy and robust interpretations across diverse real-world datasets.

3. Our experiments demonstrate that SHAPEX consistently outperforms the baseline models, delivering more accurate and reliable explanations in critical applications. To our knowledge, it is also the first approach to conduct a large-scale, occlusion-based evaluation across the entire UCR archive of 100+ datasets, establishing a more comprehensive benchmark for explanation quality.

## 2 Related Work

The field of *time series explainability* has gained increasing attention, aiming to interpret the decision process of *black-box* models applied to temporal data [14, 15]. Existing methods are commonly categorized into *in-hoc* and *post-hoc* approaches. In-hoc methods embed interpretability directly into the model, typically via transparent representations such as *Shapelets* or self-explaining architectures, as seen in TIMEVIEW [16] and VQShape [17]. In contrast, post-hoc methods generate explanations after the model has been trained and are further divided into gradient-based and perturbation-based techniques. The former includes Integrated Gradients (IG) [18] and SGT+GRAD [19], while the latter observes prediction changes under input perturbation, as in CoRTX [20] and other related work. For convenience, and following prior literature focused specifically on temporal data, we refer to post-hoc explainability methods in the time series domain as *Post-Hoc Time Series Explainability (PHTSE)*.

Several foundational studies have underscored the unique challenges of explaining time series models, such as the limitations of standard saliency techniques [15], the use of KL-divergence to quantify temporal importance [21], and the development of symbolic shapelet-based explanation rules [22]. These works laid a foundation for understanding the temporal nature of model explanations.

Building on these insights, a major line of recent work has focused on PHTSE. Unlike generic saliency methods, PHTSE methods are explicitly designed for time-dependent inputoutput mappings. Representative approaches include Dynamask [2], which learns instance-specific perturbation masks; WinIT [23], which models varying temporal dependencies via windowed perturbation; and the TIMEX family [3, 4], which ensures faithfulness through interpretable surrogate models and information bottleneck regularization. For a broader discussion of deep learning explainability methods, see Appendix B.

## 3 Methodology

Figure 2 shows the two major phases of SHAPEX. **Training:** the Shapelet Describe-and-Detect (SDD) framework learns a compact set of shapelets. **Inference:** these shapelets align segments that are (i) perturbed at the Shapelet-Driven Segment Level (SDSL) and (ii) assessed with Shapley value Attribution, producing faithful post-hoc explanations. To keep the discussion self-contained, we begin by introducing the necessary background and notation.

### 3.1 Background & Notation

We address the task of explaining time series classification models in a post-hoc manner. Given a trained black-box model that predicts class labels from time series data, our goal is to analyze its behavior by identifying which input components contribute most to the prediction.

Formally, let each input be a time series of length $T$, where each timestep $t$ has a $D$-dimensional feature vector $X_t \in \mathbb{R}^D$, forming $X = [X_1, X_2, \ldots, X_T]$. The associated label $Y$ lies in a $C$-dimensional space, $Y \in \mathbb{R}^C$. The objective of time series classification is to learn a classifier $f(\cdot)$ that maps $X$ to its predicted label $\hat{Y} = f(X)$.

To provide PHTSE, we employ an explanation method $\mathcal{E}(\cdot)$ that produces saliency scores $R \in \mathbb{R}^{T \times D}$, where each element $R_{t,d} \in [0, 1]$ quantifies the importance of the input feature $X_{t,d}$ for the predicted label $\hat{Y}$. Higher values of $R_{t,d}$ indicate stronger influence on the classification outcome. In this work, we focus on the univariate setting, i.e., $D = 1$, and omit the feature dimension $d$ for clarity; all subsequent discussions assume univariate time series unless otherwise specified. Furthermore, the evaluation of $\mathcal{E}(\cdot)$ depends on the availability of ground-truth saliency labels: when available, they serve as direct supervision or evaluation references; otherwise, occlusion-based perturbation is used to assess the quality of the generated $R$.

In the realm of PHTSE methods, perturbation-based explanation methods represent the most primary category. We briefly present the essential background here; for a more detailed discussion of perturbation paradigms, saliency granularity, see Appendix A.

The concept of *shapelets* has played a central role in time series interpretability. It was originally introduced as short, class-discriminative subsequences directly extracted from raw time series [24].

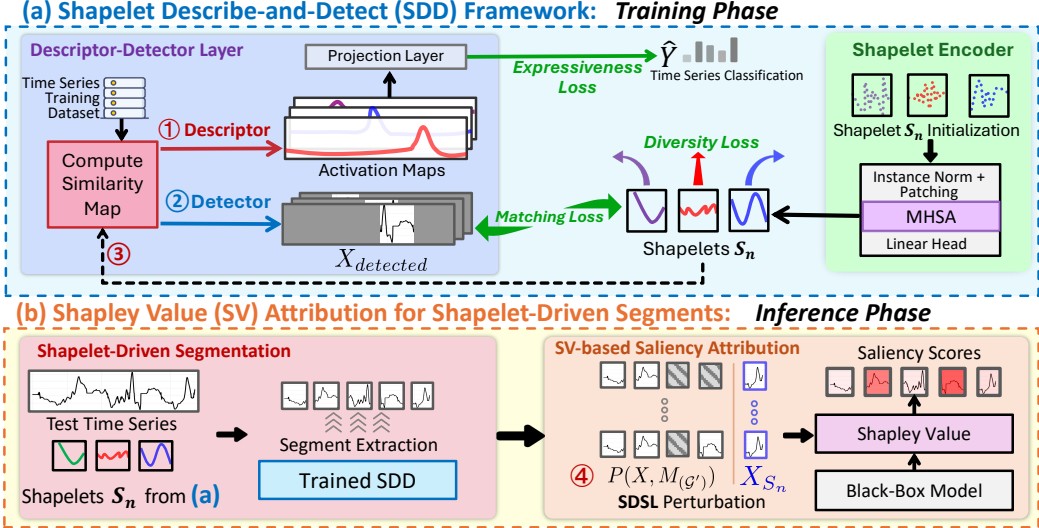

Figure 2: Overview of SHAPEX. In the training phase, given time series from dataset and the initialized shapelets, ① *descriptor* generates activation maps by ③ taking shapelets as the 1D convolution kernel, demonstrating to which extent the input time series aligns with each shapelet. Meanwile, ② *detector* locates key subsequences for the morphological matching of shapelets. Simultaneously, the shapelets are optimized using three loss functions. In inference, the learned shapelets first trigger a shapelet-driven segmentation of the test series; the resulting segments are next perturbed by SDSL to obtain ④ $P(X, M_{\mathcal{G}'})$. Finally, Shapley value is applied to compute the final saliency scores.

These subsequences served as intuitive and interpretable building blocks, revealing characteristic temporal patterns most indicative of specific classes. Under this classical definition, each shapelet corresponded to an explicit segment of the input series and was closely tied to a particular class identity. Over time, however, this notion has evolved into a more flexible and abstract representation. Modern approaches no longer require shapelets to be literal subsequences of the data; instead, they can be viewed as prototypical temporal patterns learned through optimization or representation learning [25, 26, 27, 7]. These learned shapelets are typically not class-specific but rather aim to capture generalizable and semantically meaningful features shared across the dataset.

### 3.2 Shapelet Describe-and-Detect (SDD) Framework

This section introduces the SDD framework used during training to learn a compact set of discriminative shapelets for segmenting time series into interpretable units.

Inspired by the Describe-and-Detect paradigm from image anomaly detection [28], we adapt this idea to time series by designing shapelets that both activate strongly on discriminative regions and serve as semantic prototypes. The SDD framework consists of a *descriptor-detector* layer to localize high-activation regions, a shapelet encoder to enhance representation quality, and a multi-objective training strategy that promotes classification relevance and diversity. In contrast to prior methods that rely on large shapelet banks, SHAPEX emphasizes conciseness and discriminativeness through joint optimization.

***Descriptor-Detector* Layer.** To identify regions in the input time series that align with key patterns, we apply a two-step matching mechanism consisting of a *descriptor* and a *detector*.

The *descriptor* uses a set of shapelets $\mathcal{S} = \{S_1, S_2, \ldots, S_N\}$, where each $S_n \in \mathbb{R}^L$ is a fixed-length prototype, and $N$ is the number of shapelets. For an input $X \in \mathbb{R}^T$, we compute a similarity map $I \in \mathbb{R}^{T \times N}$ using valid-width 1D convolution:

$$I = X * \mathcal{S} + b, \tag{1}$$

where $*$ denotes 1D convolution with same padding (i.e., zero padding on both sides), and $b$ is a learnable bias term. Each $I_{t,n}$ represents the similarity between shapelet $S_n$ and the local subsequence centered at position $t$.

We then apply a softmax across the shapelet dimension at each timestep to obtain an activation map $A \in \mathbb{R}^{T \times N}$:

$$A_{t,n} = \frac{e^{I_{t,n}}}{\sum_{m=1}^{N} e^{I_{t,m}}}. \tag{2}$$

Here, $A_{t,n}$ reflects the normalized alignment strength between $S_n$ and $X$ at time $t$.

The *detector* identifies the peak activation position $t_n^* = \arg\max_t A_{t,n}$ and extracts a subsequence of length $L$ centered at $t_n^*$:

$$X_n^{\text{detected}} = X[t_n^* - \lfloor L/2 \rfloor : t_n^* + \lfloor L/2 \rfloor]. \tag{3}$$

This segment $X_n^{\text{detected}} \in \mathbb{R}^L$ captures the region in $X$ most strongly aligned with shapelet $S_n$, forming the basis for downstream segmentation and attribution.

**Shapelet Encoder and Training Losses.** We apply a lightweight patch-based encoder to refine the temporal structure of each shapelet; see Appendix C for details.

To jointly optimize shapelet quality, we define a composite loss over training samples $(X, Y)$, targeting three objectives: expressiveness, alignment accuracy, and diversity. Let $\mathcal{S} = \{S_1, S_2, \ldots, S_N\}$ denote the set of learnable shapelets. The expressiveness loss encourages shapelets to be discriminative with respect to the target label. For each training mini-batch of size $K$, we obtain the predicted probabilities $\hat{Y}_i$ for each input by applying a projection layer to its activation map $A$. The expressiveness loss is defined as:

$$\mathcal{L}_{\text{cls}} = -\sum_{i=1}^{K} \left( Y_i \log \hat{Y}_i + (1 - Y_i) \log(1 - \hat{Y}_i) \right). \tag{4}$$

The matching loss promotes alignment between each $S_n \in \mathcal{S}$ and its most responsive segment $X_n^{\text{detected}}$, encouraging each shapelet to closely match the local temporal shape of the input. The diversity loss encourages non-redundancy by penalizing shapelet pairs with high similarity, promoting angular diversity and preventing redundant patterns [29]. The two losses are defined as:

$$\mathcal{L}_{\text{match}} = \sum_{n=1}^{N} d(S_n, X_n^{\text{detected}}), \qquad \mathcal{L}_{\text{div}} = \sum_{i=1}^{N} \sum_{j=i+1}^{N} \max\left(0, \delta - \text{sim}(S_i, S_j)\right), \tag{5}$$

where $d(\cdot)$ denotes the Euclidean distance, $\text{sim}(\cdot, \cdot)$ is the cosine similarity, and $\delta$ is a distance margin. The total objective is a weighted combination of the above components:

$$\mathcal{L} = \mathcal{L}_{\text{cls}} + \lambda_{\text{match}} \mathcal{L}_{\text{match}} + \lambda_{\text{div}} \mathcal{L}_{\text{div}}, \tag{6}$$

where $\lambda_{\text{match}}$ and $\lambda_{\text{div}}$ balance alignment and diversity.

### 3.3 Shapley Value Attribution for Shapelet-Driven Segments

To perform post-hoc time series explanation, SHAPEX evaluates the importance of shapelet-aligned segments through Shapley value analysis. These segments correspond to regions aligned with the learned shapelets. Since this framework relies on marginal contributions over different coalitions, we simulate segment inclusion or exclusion via perturbation: retaining a segment corresponds to including it in the coalition, while perturbing it simulates its removal. This enables the computation of saliency scores reflecting each segments contribution to the model output.

**Shapelet-Driven Segmentation.** To extract these segments, SHAPEX utilizes the shapelets learned during training to extract interpretable segments that correspond to key morphological patterns. These segments are derived from shapelet activation maps computed on the test input, enabling a structured and semantically meaningful decomposition of the time series.

Specifically, for a time series $X \in \mathbb{R}^T$ from the test set and each learned shapelet $S_n \in \mathcal{S} = \{S_1, \ldots, S_N\}$, we compute the activation map $A_n \in \mathbb{R}^T$ using the trained SDD from Section 3.2, indicating alignment strength between $X$ and $S_n$ across timesteps.

To extract high-activation regions, we apply a threshold $\Omega$ to obtain:

$$\mathcal{T}_{(S_n)} = \{\, t \in \{1, \ldots, T\} \mid A_{n,t} > \Omega \,\}. \tag{7}$$

The corresponding segment is:

$$X_{(S_n)} = \{ X_t \mid t \in \mathcal{T}_{(S_n)} \}, \tag{8}$$

capturing the region in $X$ that is morphologically aligned with shapelet $S_n$.

Let $\mathcal{G} = \{1, 2, \ldots, N\}$ index the set of shapelet-aligned segments $\{X_{(S_n)}\}_{n=1}^{N}$, each corresponding to a shapelet $S_n \in \mathcal{S}$. For any subset $\mathcal{G}' \subseteq \mathcal{G}$, the union of activated timesteps is defined as:

$$\mathcal{T}_{(\mathcal{G}')} = \bigcup_{n \in \mathcal{G}'} \mathcal{T}_{(S_n)}. \tag{9}$$

These shapelet-driven segments serve as explanation units in the Shapley value analysis that follows.

**Shapelet-driven Segment-Level (SDSL) Perturbation.** We define a binary perturbation mask $M_{(\mathcal{G}')} \in \mathbb{R}^T$ for any subset of segment indices $\mathcal{G}' \subseteq \mathcal{G} = \{1, \ldots, N\}$:

$$M_{(\mathcal{G}')} = \begin{cases} 1, & t \in \mathcal{T}_{(\mathcal{G}')} \\ 0, & t \in \mathcal{T} \setminus \mathcal{T}_{(\mathcal{G}')}, \end{cases} \tag{10}$$

where the perturbation function defined in Equation 14 is instantiated as $P(X_t, M_{(\mathcal{G}')})$. However, the current baseline valuing approach, whether zero-filling or averaging, often introduces abrupt changes at boundaries. These changes may prevent the value function in Shapley value from fully eliminating the influence of segments outside the coalition. To address this, we propose a linear perturbation, where the baseline value $B_t$ is set as:

$$B_t = X_{t_{\text{start}}} + \frac{t - t_{\text{start}}}{t_{\text{end}} - t_{\text{start}}} (X_{t_{\text{end}}} - X_{t_{\text{start}}}), \quad t \in [t_{\text{start}}, t_{\text{end}}], \tag{11}$$

where $X_{t_{\text{start}}}, X_{t_{\text{end}}}$ are the values at the boundaries of the perturbed region. This design isolates the contribution of $X_{(\mathcal{G}')}$ and enables Shapley value computation. Intuitively, this linear interpolation creates a smooth transition between the two boundaries, preventing artificial discontinuities when a segment is masked.

**Shapley Value Computation.** We compute the Shapley value $\phi_n$ for each segment $X_{(S_n)}$ (indexed by $n \in \mathcal{G}$) as:

$$\phi_n = \sum_{\mathcal{G}' \subseteq \mathcal{G} \setminus \{n\}} \frac{|\mathcal{G}'|! \, (|\mathcal{G}| - |\mathcal{G}'| - 1)!}{|\mathcal{G}|!} \left[ f\left( P\left( X, M_{(\mathcal{G}' \cup \{n\})} \right) \right) - f\left( P\left( X, M_{(\mathcal{G}')} \right) \right) \right], \tag{12}$$

where $X_{t_{\text{start}}}$ and $X_{t_{\text{end}}}$ are the values at the boundaries of the perturbed region, and $P(X, M_{(\mathcal{G}')})$ denotes the perturbed input defined in Equation 14. The coefficient gives the probability that segment $n$ is added after the subset $\mathcal{G}'$ in a random ordering of all segments, and $f(\cdot)$ denotes the black-box classifier.

To further reduce the complexity in computation and based on the assumption of temporal dependency, we apply a temporary-relational subset extract strategy, which is to restrict the range of coalitions in Shapley value to a subset of segments that are directly or indirectly connected to the current $X_{(S_n)}$. This modification reduces the computational complexity of each segments Shapley value from $O(N!)$ to $O(N)$. Ultimately, the saliency score is computable using Equation 19 (see Appendix A for the detailed formulation).

## 4    ShapeX as Approximate Causal Attribution

In this section, we present a theoretical analysis of SHAPEX as a method for approximate causal attribution in time series models.

Most post-hoc explainers for time series models merely expose **correlations**: they estimate the associations between input regions and model predictions, often by training proxy models or computing

correlation-based importance scores. However, these methods do not guarantee that the highlighted regions are **causal drivers** of the model's predictions. In contrast, SHAPEX is designed to provide **model-level causal insight** by explicitly conducting causal intervention [30]. Specifically, SDSL perturbation interprets the actions of "keeping vs. masking a segment" as an intervention in the models input space, while shapelet-driven segmentation isolates semantically coherent subsequences for calculating Shapley value. By framing perturbations as interventions, SHAPEX moves beyond correlational explanations to provide causal attributions that are more robust to confounding noise, more stable across re-training, and better aligned with practitioners needs in sensitive domains such as medical triage.

To formalize the causal semantics of SHAPEX, we demonstrate that its attribution mechanism approximates the concept of the **CATE** [31, 32]. The CATE quantifies the expected change in an outcome resulting from a specific intervention, conditioned on a given context. In our setting, the outcome is the model prediction $f(X)$, the intervention is whether a shapelet-aligned segment $X_{(S_n)}$ is retained or masked, and the context is defined by the remaining segments indexed by $\mathcal{G}' \subseteq \mathcal{G} \setminus \{n\}$, where $\mathcal{G} = \{1, \ldots, N\}$ is the set of all segment indices. Shapelet-driven segmentation naturally provides localized and semantically coherent regions, making it particularly well-suited for modeling such conditional interventions. Importantly, the resulting CATE in this setting is defined with respect to the **model prediction**, not the true outcome $Y$. Hence, we refer to this quantity as the **model-level CATE**, denoted by $\tau^{\mathrm{model}}$, measuring the causal strengths of $X_{(S_n)}$ on $f(X)$. We formalize this relationship in the following proposition:

**Proposition 1** (Shapley Value as Model-Level CATE). *Let $f(X)$ denote the models prediction for input $X$, and let $X_{(S_n)}$ denote the segment aligned with shapelet $n \in \mathcal{G}$, where $\mathcal{G} = \{1, \ldots, N\}$ is the index set of all segments. Then, under the intervention defined by retaining or masking segment $X_{(S_n)}$, the Shapley value $\phi_n$ computed as*

$$\phi_n = \mathbb{E}\mathcal{G} \left[ f(X_{\mathcal{G}' \cup \{n\}}) - f(X_{\mathcal{G}}) \right] \tag{13}$$

*is equivalent to the model-level CATE $\tau_n^{model}$, conditioned on context $\mathcal{G}' \subseteq \mathcal{G} \setminus \{n\}$.*

A detailed proof is provided in Appendix D. This equality holds exactly when all coalitions $\mathcal{G}$ are exhaustively enumerated. Under subset sampling, the estimator remains unbiased given a valid surrogate randomization mechanism.

We further conduct a case study in Figure 3 to underline this mechanism, see Appendix G.3 for details: TIMEX++-based maps scatter importance across many timesteps, hinting only at a loose association with the outcome. SHAPEX, however, concentrates saliency on the growth plate transition region, the true biomechanical determinant of class labels, providing visually plausible evidence of a causal structure.

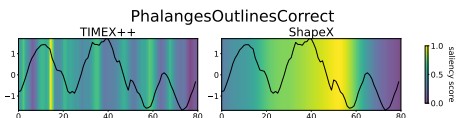

Figure 3: The saliency scores generated by SHAPEX and TIMEX++ on PhalangesOutlineCorrect dataset.

## 5 Experiments

The explanatory effectiveness of SHAPEX is assessed across four synthetic datasets and a comprehensive suite of real-world datasets, considering two different perspectives: (1) For datasets with ground-truth saliency scores, we conduct saliency score evaluation experiments. (2) For datasets without ground-truth saliency scores, we perform occlusion experiments [2]. The best and second-best results in all experiments are highlighted in bold and underlined, respectively. In terms of the black-box models to be explained, we select four of the most widely used time series classification models: Transformer [33], LSTM [34], CNN [35] and MultiRocket [36]. The experimental results presented in the main body are all based on vanilla Transformer [33] as black-box models. Further experimental details can be found in Appendix E.

**Datasets.** We evaluate on both synthetic and real-world time series. The synthetic data includes four motif-based binary classification datasets: **(i) MCC-E**, **(ii) MTC-L**, **(iii) MCC-L**, and **(iv) MTC-E**, following [37]. Each sample is annotated with ground-truth saliency for evaluating explanation quality. For real-world data, we use the **ECG** dataset [38], which similarly provides ground-truth labels, and the full **UCR Archive** [39], comprising over 100 univariate classification datasets across diverse domains. Further details are provided in Appendix E.1.

Table 1: Saliency evaluation on synthetic datasets using Transformers as black-box model. Dark blue marks better values.

| METHOD | MCC-E | | | MCC-H | | |
|---|---|---|---|---|---|---|
| | AUPRC | AUP | AUR | AUPRC | AUP | AUR |
| IG [18] | 0.3630±0.0052 | 0.4825±0.0077 | 0.4671±0.0053 | 0.4363±0.0058 | 0.5100±0.0061 | 0.5793±0.0053 |
| DYNAMASK [2] | 0.3251±0.0046 | 0.4722±0.0098 | 0.0711±0.0026 | 0.3506±0.0034 | 0.1832±0.0023 | 0.1574±0.0034 |
| WINIT [23] | 0.1675±0.0044 | 0.1575±0.0113 | 0.3861±0.0066 | 0.1482±0.0036 | 0.0974±0.0054 | 0.4442±0.0073 |
| CoRTX [20] | 0.4762±0.0055 | 0.5238±0.0174 | 0.4589±0.0157 | 0.6195±0.0036 | 0.5358±0.0159 | 0.5428±0.0033 |
| MILLET [40] | 0.1730±0.0057 | 0.2178±0.0071 | 0.1641±0.0072 | 0.3016±0.0046 | 0.2741±0.0082 | 0.2951±0.0034 |
| TIMEX [3] | 0.3691±0.0042 | 0.3207±0.0038 | 0.6313±0.0033 | 0.4436±0.0029 | 0.6668±0.0055 | 0.3649±0.0031 |
| TIMEX++[4] | 0.3676±0.0038 | 0.4998±0.0064 | 0.2801±0.0037 | 0.6393±0.0041 | 0.6411±0.0038 | 0.4879±0.0015 |
| SHAPEX_SF | 0.1832±0.0024 | 0.1709±0.0061 | 0.2215±0.0043 | 0.2515±0.0061 | 0.1511±0.0061 | 0.7210±0.0057 |
| SHAPEX | 0.6407±0.0036 | 0.5614±0.0076 | 0.3679±0.0050 | 0.8113±0.0013 | 0.6838±0.0054 | 0.7431±0.0055 |

| METHOD | MTC-E | | | MTC-H | | |
|---|---|---|---|---|---|---|
| | AUPRC | AUP | AUR | AUPRC | AUP | AUR |
| IG [18] | 0.1467±0.0011 | 0.1468±0.0027 | 0.5402±0.0041 | 0.3096±0.0030 | 0.5317±0.0061 | 0.4963±0.0059 |
| DYNAMASK [2] | 0.1388±0.0006 | 0.1010±0.0020 | 0.2796±0.0043 | 0.2596±0.0022 | 0.4955±0.0074 | 0.1894±0.0034 |
| WINIT [23] | 0.1407±0.0029 | 0.0914±0.0053 | 0.0053±0.0062 | 0.1504±0.0038 | 0.0914±0.0045 | 0.4624±0.0060 |
| CoRTX [20] | 0.1875±0.0061 | 0.1749±0.0096 | 0.5259±0.0171 | 0.2428±0.0101 | 0.2405±0.0143 | 0.5430±0.0056 |
| MILLET [40] | 0.1419±0.0053 | 0.1103±0.0037 | 0.1550±0.0153 | 0.2071±0.0109 | 0.2312±0.0025 | 0.3518±0.0010 |
| TIMEX [3] | 0.2479±0.0025 | 0.6301±0.0070 | 0.0935±0.0009 | 0.3799±0.0016 | 0.8890±0.0023 | 0.1542±0.0010 |
| TIMEX++[4] | 0.2424±0.0021 | 0.4906±0.0064 | 0.2632±0.0031 | 0.3903±0.0025 | 0.8996±0.0014 | 0.1448±0.0012 |
| SHAPEX_SF | 0.3569±0.0032 | 0.1520±0.0036 | 0.4611±0.0015 | 0.3852±0.0124 | 0.7350±0.0046 | 0.1384±0.0010 |
| SHAPEX | 0.6100±0.0048 | 0.3962±0.0067 | 0.5472±0.0082 | 0.6792±0.0014 | 0.4255±0.0024 | 0.9019±0.0041 |

Table 2: Saliency score on the ECG dataset.

| METHOD | AUPRC | AUP | AUR |
|---|---|---|---|
| IG | 0.4182±0.0014 | 0.5949±0.0023 | 0.3204±0.0012 |
| DYNAMASK | 0.3280±0.0011 | 0.5249±0.0030 | 0.1082±0.0080 |
| WINIT | 0.3049±0.0011 | 0.4431±0.0026 | 0.3474±0.0011 |
| CoRTX | 0.3735±0.0008 | 0.4968±0.0021 | 0.3031±0.0009 |
| MILLET | 0.3017±0.0024 | 0.4721±0.0062 | 0.3098±0.0008 |
| TIMEX | 0.4721±0.0018 | 0.5663±0.0025 | 0.4457±0.0018 |
| TIMEX++ | 0.6599±0.0009 | 0.7260±0.0010 | 0.4595±0.0007 |
| SHAPEX_SF | 0.4723±0.0012 | 0.6851±0.0047 | 0.3274±0.0034 |
| SHAPEX | 0.7228±0.0028 | 0.8395±0.0030 | 0.6961±0.0032 |

Table 3: Ablation on ECG.

| Ablations | AUPRC | AUP | AUR |
|---|---|---|---|
| Full | 0.7225±0.0028 | 0.8399±0.0030 | 0.6958±0.0032 |
| w/o M | 0.2398±0.0022 | 0.2055±0.0019 | 0.7117±0.0041 |
| w/o D | 0.3671±0.0030 | 0.4973±0.0047 | 0.3619±0.0022 |
| w/o SE | 0.2073±0.0019 | 0.1597±0.0014 | 0.7504±0.0037 |
| w/o LIN | 0.7085±0.0030 | 0.7739±0.0036 | 0.6904±0.0032 |
| w/o SEG | 0.6975±0.0026 | 0.7321±0.0033 | 0.3847±0.0019 |

M: matching loss ; D: diversity loss ; SE: shapelet encoder
LIN: LINEAR; SEG: SEGMENT.

**Benchmarks.** The proposed methods are compared with multiple benchmark approaches (see Appendix E.2 for benchmark details and metric definitions.): *Perturbation-based methods:* **TimeX++** [4], **TIMEX** [3],**Dynamask** [2] and **CoRTX** [20]. *Gradient-based methods:* **Integrated Gradients (IG)** [18], **WinIT** [23], **MILLET** [40].

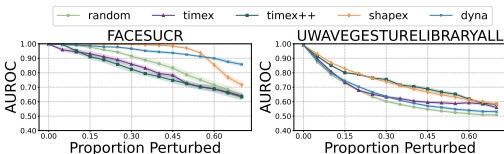

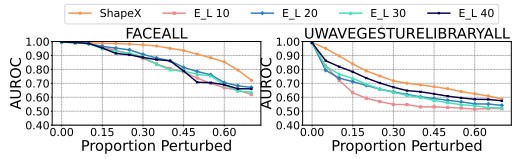

Figure 4: Occlusion results under different perturbation ratios, where higher AUROC indicates better saliency.

Figure 5: Ablation study on SHAPEX by replacing SDSL with equal-length segmentation, leading to performance drop.

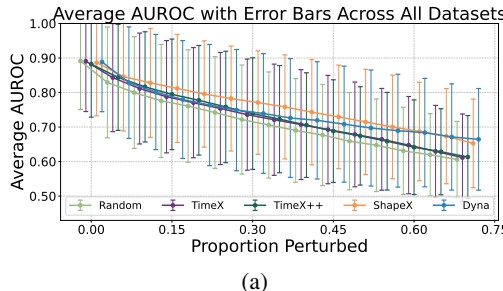

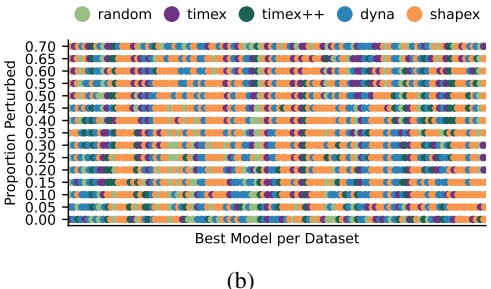

(a)

(b)

Figure 6: Occlusion evaluation results on the full UCR archive. (a) Average AUROC with standard deviation across all datasets under varying perturbation ratios. (b) Distribution of the best-performing explainability method across datasets under varying perturbation ratios. Each row corresponds to a perturbation level, and each colored dot represents the method achieving the highest performance on a specific dataset at that level.

## 5.1 Saliency Score Evaluation on Synthetic Datasets and Real-world Datasets

In this section, we directly evaluate the generated saliency score by each model on the synthetic dataset and the real-world dataset. We also report SHAPEX_SF, using ShapeFormer [6] in place of SDD (see Subsection 5.3 for details). In Table 1, SHAPEX surpasses the second-best method in AUPRC by an average of 58.12%, with the largest gain of 146.07% on MTC-E. For AUR, it achieves an average improvement of 21.09%. However, in AUP, SHAPEX shows a slight performance decline. We make the following observations: The consistent enhancement of SHAPEX in AUPRC indicates that it effectively balances the identification of complete salient sequences while maintaining high accuracy. Notably, TIMEX and TIMEX++ perform significantly worse on the MCC-E and MTC-E datasets, where the motif amplitude is equal to the baseline waveform, compared to the MCC-H and MTC-H datasets, where motifs exhibit higher amplitudes. This suggests that these methods

tend to label outliers or anomalies as key features. In contrast, SHAPEX remains stable across all datasets, demonstrating its ability to identify key time series patterns relevant to classification rather than relying on extreme amplitude variations. Table 2 shows that SHAPEX also consistently outperforms all baselines, with a notable AUR improvement in ECG dataset. To explore how different black-box classifiers affect saliency outcomes, we also conduct further evaluations using a range of models, from standard LSTM [34] and CNN [35] to the state-of-the-art MultiRocket [36]. Results are presented in Appendix F.

## 5.2 Occlusion Experiments on Real-world Datasets

We follow the occlusion protocol from [3], which perturbs the bottom-$k$ timesteps ranked by saliency scores generated from explanation methods. By progressively perturbing less important regions, we observe how quickly the model's classification accuracy degrades. A reliable explanation model should exhibit a slow and smooth performance drop—indicating that high saliency regions truly capture the key discriminative patterns, while a steep drop suggests poor alignment between the saliency map and the underlying predictive logic.

Figure 4 illustrates occlusion results on two typical UCR datasets, serving as visual examples of the evaluation process rather than highlighting best-case performance. Full occlusion results for all datasets are included in Appendix F.2. Figure 6 summarizes occlusion results across 112 datasets from the UCR archive. Subfigure (a) presents the average AUROC under different perturbation ratios, where SHAPEX consistently achieves the highest scores and exhibits the most stable degradation curve, indicating superior robustness. Subfigure (b) reports the frequency with which each method ranks first across all datasets and occlusion levels, further confirming the reliability of SHAPEX.

*To our knowledge, this is the first PHTSE work to conduct explanation evaluation over the entire UCR archive*, covering a wide range of real-world domains, sequence lengths, and class granularities. The results not only validate the strong generalization ability of SHAPEX but also demonstrate its unmatched consistency and reliability across diverse conditions.

## 5.3 Ablation Study

We conduct ablation studies on both real-world and synthetic datasets to assess the contributions of different components in SHAPEX. The results on the ECG dataset are shown in Table 3, while additional results on synthetic datasets are provided in Appendix F. As the trends are consistent, we focus our discussion on the ECG setting.

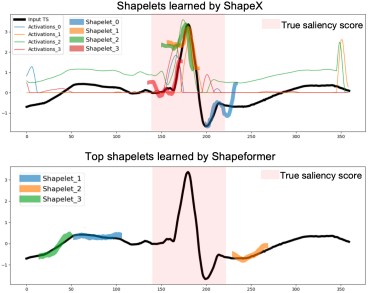

We first evaluate SHAPEX_SF, a variant that replaces the proposed SDD module with ShapeFormer [6] for shapelet learning, while keeping the downstream explanation pipeline unchanged. Despite ShapeFormer's ability to model local patterns, this variant performs worse in saliency evaluation, highlighting that shapelets must be learned in an explanation-aware manner to be effective for post-hoc interpretation. Next, we ab-

Figure 7: Visualization of learned shapelets in ECG dataset.

late key components of the SDD module. Specifically, we remove the two loss functions that guide morphological learningthe matching loss and the diversity lossas well as the shapelet encoder. W/O LINEAR replaces the linear perturbation with a zero-mask perturbation, and W/O SEGMENT directly uses the shapelet activation map as the saliency score without segment-based aggregation.

From the results, we observe that the loss components have the most significant impact on model performance, confirming that the learned shapelets must be well-aligned with the input, temporally smooth, and morphologically diverse. Furthermore, the sharp drop in AUR for W/O SEGMENT validates the necessity of segment-level aggregation for capturing complete key subsequences.

To assess the impact of shapelet-driven segmentation, we perform an ablation study involving occlusion experiments on two real-world datasets. Specifically, we replace the SDSL perturbation in SHAPEX with equally spaced segments of length $N$, denoted as $E\_L N$. The results are shown in

Figure 5. SHAPEX significantly outperforms other variants. These results indicate that the effectiveness of SHAPEX is fundamentally dependent on SDSL perturbation.

## 5.4 Shapelet Visualization and Case Study

**Visualization.** ShapeFormer [6] is a leading shapelet learning method. We visualize and compare the shapelets generated by ShapeFormer and SHAPEX, as shown in Figure 7. In accordance with the initial configuration of ShapeFormer, we choose the shapelets it generates that have the highest PSD scores within the dataset. The results show that ShapeFormers shapelets primarily match subsequences within the time series, while SHAPEX learns shapelets critical for classification. Specifically, SHAPEX's shapelets align with saliency labels, emphasizing their relevance to classification. This indicates that only SHAPEX's shapelets effectively support downstream explanations.

**Case Study.** Additionally, case studies on three real-world datasets (Appendix G.3) further validate SHAPEXs causal fidelity and interpretive precision. In each case, SHAPEX produces saliency distributions that align closely with the ground-truth causal regions verified by domain knowledge or prior annotations. For example, on the PhalangesOutlinesCorrect dataset, the model accurately highlights the transitional growth plate region that determines bone maturity, while other baselines yield dispersed and noisy saliency maps without semantic correspondence. Similarly, for the FaceAll dataset, SHAPEX concentrates attention around facial boundaries and expression-related areaskey morphological cues for class discrimination, demonstrating its ability to capture meaningful shape transitions rather than texture noise. Finally, in motion-related datasets such as UWaveGestureLibraryAll [41], SHAPEX highlights subsequences with decreasing acceleration, which correspond to the transitional phases of gestures such as sharp turns or circular motions. Overall, SHAPEX identifies truly causal segments, while other methods focus on random or uninformative regions.

## 6 Conclusion

SHAPEX introduces an innovative shapelet-driven approach for explaining time series classification. By leveraging the SDD framework, SHAPEX effectively learns representative and diverse shapelets, integrating Shapley value to assess their contribution to classification outcomes. Experimental results demonstrate that SHAPEX not only enhances precision in identifying key subsequences but also provides explanations with stronger causal fidelity. Its effectiveness across synthetic and real-world datasets highlights its potential for improving interpretability in critical applications such as healthcare and finance.

However, several limitations remain. The framework involves a few user-defined hyperparameters, such as the number and length of shapelets and the threshold that determines their selection. These parameters control the granularity and diversity of extracted shapelets, which in turn influence both interpretability and quantitative performance. While they offer flexibility across tasks, they may also require dataset-specific tuning and manual calibration to achieve optimal results, limiting the methods plug-and-play usability on unseen domains.

Meanwhile, SHAPEX relies on a separate training phase to learn shapelets and optimize the SDD module, which introduces additional computational overhead compared to purely gradient-based methods. This two-stage design ensures interpretability and robustness but sacrifices some efficiency and adaptability. Future work could address these limitations through automated hyperparameter selection, lightweight or joint-training strategies, and generalized formulations for multivariate and irregular time series.

## Acknowledgments

S. Pan was partially funded by Australian Research Council (ARC) under grants FT210100097 and DP240101547 and the CSIRO  National Science Foundation (US) AI Research Collaboration Program. This work was also partially supported by the NVIDIA Academic Grant in Higher Education and the NVIDIA Developer Program.

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

# A Supplementary Background: Perturbation Paradigms and Shapelets

## A.1 Perturbation-based Explanation Methods

Perturbation-based methods derive the saliency of input features by analyzing how perturbations to those features affect the models output. Let $P(X, M)$ represent a perturbation function that modifies the input $X$ using a mask $M = [M_1, M_2, \ldots, M_T]$, where $M_t \in [0, 1]$. The perturbed $X$ is defined as:

$$X'_t := P(X_t, M_t) = M_t \cdot X_t + (1 - M_t) \cdot B_t, \tag{14}$$

where $B_t$ denotes a baseline value (e.g., zero, mean, or noise). A general formulation of perturbation-based explanation methods to evaluate saliency is given by:

$$R \propto \mathbb{E}_{M \sim \mu}\Big[ G\left(f(X) - f(P(X, M))\right) \Big], \tag{15}$$

where $\mu$ is the distribution of the perturbation mask, and $G(\cdot)$ represents some transformation function. Two primary subclasses can be utilized to instantiate this method: the *Optimized Mask* and *Fixed Mask* methods.

### A.1.1 Two Primary Subclasses of Perturbation-Based Explanation Methods

**Optimized Mask Perturbation Methods.** In these methods, mask $M$ is treated as a learnable parameter. The optimal mask $M^*$ is obtained by solving an optimization problem [2, 3, 4], and the optimized mask $M^*$ directly represents the saliency score, with $R_t = M_t^*$.

**Fixed Mask Perturbation Methods.** wherein $M$ is a predefined, fixed mask (e.g., masking one time step or a segment). The saliency score $R$ is defined as:

$$R = \mathbb{E}_{M \sim \mathcal{M}}\Big[\big|f(X) - f(P(X, M))\big|\Big], \tag{16}$$

where $\mathcal{M}$ is the distribution of the mask $M$, and $|\cdot|$ denotes a norm, typically $L_1$ or $L_2$, to measure the change in the models output.

Within this field, the Shapley value is frequently employed to determine the marginal contribution of each unmasked element (as a player in Shapley value) to a model's behavior, which can then be converted to the saliency score. This branch primarily focuses on Perturbing equal-length segments [42, 43]. Refer to Appendix A.2 for additional information on using equal-length perturbed segments to compute Shapley value.

## A.2 Applying Shapley Value to Fixed Mask Perturbation

Shapley value [13] offers a principled approach to measure the importance of each segment by averaging their marginal contributions across all possible subsets of segment indices. In fixed mask perturbation methods, segments of the time series are treated as participants, and the Shapley value is computed using the model's output as the value function. The saliency scores derived from Shapley value provide interpretable and consistent importance measures for segments or individual time steps within the time series.

Under the fixed mask perturbation framework, suppose the time series $X = \{X_1, \ldots, X_T\}$ is equally divided into $n$ segments $\{E_1, E_2, \ldots, E_n\}$. Each segment $E_i$ acts as a "participant" that can either be retained (unmasked) or masked with a baseline value. Define the mask $M \in \{0, 1\}^n$, where $M_i = 1$ indicates that segment $E_i$ is retained, and $M_i = 0$ indicates it is masked. For a subset of segment indices $u \subseteq \mathcal{N}$, the value function is defined as:

$$v(u) = f\left(P\left(X, M_u\right)\right), \tag{17}$$

where $M_u$ is a mask that retains segments with indices in $u$ and masks all other segments.

The Shapley value $\phi_i$ for each segment $E_i$ is then computed as:

$$\phi_i = \sum_{u \subseteq \mathcal{N} \setminus \{i\}} \frac{|u|!\,(n - |u| - 1)!}{n!} \left[f\left(P\left(X, M_{u \cup \{i\}}\right)\right) - f\left(P\left(X, M_u\right)\right)\right]. \tag{18}$$

This equation captures the average marginal contribution of segment $E_i$ across all possible subsets $u$ of segment indices that do not include $i$. The resulting $\phi_i$ serves as the saliency score for segment $E_i$. To obtain saliency scores for individual time steps $t$, distribute the segment-level Shapley value uniformly across the time steps within each segment:

$$R_t = \frac{|\phi_i|}{|E_i|}, \quad \text{for } t \in E_i. \tag{19}$$

As the general form of fixed mask perturbation methods is given by:

$$R_t = \mathbb{E}_{M \sim \mathcal{M}} \left[ |f(X) - f(P(X, M))| \right],$$

where $\mathcal{M}$ is the distribution over masks. By leveraging the Shapley value, the distribution $\mathcal{M}$ can be interpreted as considering all possible subsets of segment indices with equal probability. Thus, the Shapley value-based saliency scores $\phi_i$ provide a theoretically grounded method to estimate the importance of each segment in the fixed mask perturbation framework.

### A.3 Timestep-level and Segment-level Perturbation

Previously, we categorized perturbation-based explanation methods [4, 3, 2, 44, 42, 43] based on whether the mask was fixed. From another perspective, we classify them by the minimal element in the perturbation mask: **timestep-level** and **segment-level**, indicating whether the mask is applied to individual timesteps or segments. All the optimized mask methods are timestep-level, while most fixed mask methods are segment-level.

However, timestep-level perturbation directly disrupts the local dynamic dependencies, while equal length segment-level perturbation may disrupt the integrity of critical subsequences by splitting them across adjacent segments or combining unrelated subsequences into a single segment, refer to Figure 1. This can lead to fragmentation of meaningful patterns and generation of spurious segments, yielding suboptimal explanations.

## B    Additional Related Works and Discussion

**Explainable Artificial Intelligence.** Explainable Artificial Intelligence (XAI) aims to make AI systems' decisions understandable and auditable [45]. The growing complexity of deep learning models necessitates XAI methods to address safety, fairness, and accountability concerns. XAI is generally categorized into in-hoc and post-hoc explainability methods.

In-hoc explainability methods are inherently interpretable by design. In the field of computer vision, in-hoc methods include such as interpretable representation learning [46, 47], and model architectures with intrinsic interpretability [48]. Mean while, transparent generative models, such as interpretable Generative Adversarial Networks (GANs), are also used to provide visual explanations for image generation processes [49]. In natural language processing, in-hoc methods leverage interpretable attention mechanisms to reveal the relationship between input text and output predictions [50]. For tabular data, in-hoc methods often rely on comprehensible models like decision trees or linear models [51].

Post-hoc explainability methods provide explanations for complex models after training, making them applicable to various data modalities. Among these, Shapley value, derived from game theory, is widely used in XAI to quantify the contribution of each feature to model predictions [24]. The advantage of Shapley value lies in its adherence to principles of fairness, equitability, and consistency. Methods based on Shapley value, such as SHAP (SHapley Additive exPlanations), approximate Shapley value to offer both global and local explanations for complex models [52]. In computer vision, SHAP identifies the pixels or regions in an image that have the greatest influence on classification outcomes, helping to understand the key areas the model focuses on [53]. In natural language processing, Shapley value are used to evaluate the contribution of each word to text classification or generation tasks, revealing the basis for the model's decisions [54].

Besides Shapley value, post-hoc methods include techniques such as LIME [55], Grad-CAM [56], Layer-wise Relevance Propagation (LRP) [57], and Integrated Gradients [18]. These methods pro-

vide fine-grained explanations without altering the original model structure, aiding in the understanding of feature extraction and decision-making processes.

Through these methods, the application of XAI across various data modalities not only enhances the transparency and interpretability of models but also strengthens user trust in AI systems, laying a solid foundation for their deployment and adoption in real-world applications.

**Time Series Explainability.** Explainability methods for time series models are broadly categorized into in-hoc and ad-hoc approaches. Early survey and benchmarking studies have established the foundation of this field. Theissler *et al.* [58] provided a comprehensive taxonomy of explainable AI for time series classification, grouping methods into time-point-, subsequence-, and instance-based explanations, while Ismail *et al.* [15] systematically benchmarked saliency-based interpretability techniques, revealing their limitations in capturing temporal importance.

In-hoc methods embed interpretability directly into the model, typically via transparent representations such as *Shapelets* or self-explaining architectures. For instance, TIMEVIEW adopts a top-down transparency framework to understand both high-level trends and low-level properties, leveraging static features and visualization tools for interpretability [16], while VQShape introduces a novel representation learning approach based on vector quantization, enabling shapelet-based interpretable classification [17]. Spinnato *et al.* [59] further advanced this direction by developing a subsequence-based explainer capable of generating saliency, instance-level, and rule-based explanations for any black-box time series classifier.

Ad-hoc methods are typically divided into two main categories. *Gradient-Based Methods* are mainly derived from IG [18]. For instance, SGT+GRAD refines gradient-based saliency maps by iteratively masking noisy gradient features to improve interpretability [19]. *Perturbation-Based Methods* derive the saliency by observing the effect of perturbation. Tonekaboni *et al.* [60] proposed FIT, a framework quantifying instance-wise feature importance via KL-divergence to identify when and where a models decision changes over time. Dynamask creates dynamic perturbation masks to yield feature importance scores specific to each instance [2]. CoRTX enhances real-time model explanations using contrastive learning to reduce reliance on labeled explanation data [20]. WinIT improves perturbation-based explainability by explicitly accounting for temporal dependence and varying feature importance over time [23]. TIMEX learns an interpretable surrogate model that maintains consistency with a pretrained models behavior to ensure faithfulness [3]. TIMEX++ builds on the information bottleneck principle to generate label-preserving explanations while mitigating distributional shifts [4].

The principal distinction between these methods lies in their interpretability strategies: in-hoc approaches seek intrinsic transparency of the model itself, whereas ad-hoc strategies aim to provide explanations for the behaviors of the model, with perturbation-based techniques currently prevailing in the domain.

## C Shapelet Econder

The shapelets $\mathcal{S}$ used in *descriptor* and *detector* are not fixed but learned jointly with the model. To ensure their internal consistency and expressive capacity, we introduce a dedicated encoder that models their temporal structure. Each shapelet is segmented into local patches, which are embedded with positional encoding as in [61]. A multi-head self-attention mechanism is then applied to model temporal dependencies across patches, promoting smoothness and internal consistency in shapelet representations. This encoder helps shapelets act as both expressive and stable morphological prototypes.

We first divide the input shapelet sequence $s_m \in \mathbb{R}^L$ into several equal-length patches, where each patch captures a continuous segment of temporal features and is mapped into a representation $p_i$ through a linear transformation combined with positional encoding. Specifically, the representation of each patch is given by

$$p_i = \mathrm{Linear}(s_i) + PE(i), \tag{20}$$

where $\mathrm{Linear}(\cdot)$ denotes the linear mapping operation and $PE(i)$ is the positional encoding that preserves the position information of the patch. Next, we apply a multi-head attention module to

perform information exchange and fusion among all patch representations. For the set of patch representations $p_1, p_2, \ldots, p_N$ obtained after the linear transformation, the attention is computed using the following formula:

$$\text{Attention}(Q, K, V) = \text{softmax}\left(\frac{QK^T}{\sqrt{d_k}}\right)V, \tag{21}$$

where $Q$, $K$, and $V$ are the query, key, and value matrices, respectively, obtained through linear projections of $p_i$. The multi-head attention mechanism utilizes multiple sets of these linear projections to capture different feature patterns in parallel, which can be expressed as

$$\text{MultiHead}(Q, K, V) = W^O \left[\text{head}_1; \text{head}_2; \ldots; \text{head}_h\right], \tag{22}$$

with each individual attention head computed as

$$\text{head}_i = \text{Attention}(QW_i^Q, KW_i^K, VW_i^V), \tag{23}$$

where index $i$ refers to different attention heads. This design enables the Shapelet Encoder to not only preserve the fine-grained information of local patches but also to integrate global temporal dependencies through multi-head attention, thus aiding in the learning of more precise and stable shapelets.

## D  Theoretical Justification for Approximate Causal Attribution

### D.1  Theoretical Analysis

In this section, we provide a theoretical justification for interpreting the Shapley value computed by SHAPEX as an approximation of the model-level CATE. We begin by introducing the necessary notation. Let $z \in \{0, 1\}$ indicate whether a segment indexed by $n$ is retained ($z = 1$) or masked ($z = 0$), and let $\mathcal{G}' \subseteq \mathcal{G} \setminus \{n\}$ denote a coalition of other retained segments, where $\mathcal{G} = \{1, \ldots, N\}$ is the index set of all shapelet-aligned segments in the input.

We define the potential model outcomes under each intervention as follows:

$$Y(1, \mathcal{G}') = f(X_{\mathcal{G}' \cup \{n\}}), \tag{24}$$
$$Y(0, \mathcal{G}') = f(X_{\mathcal{G}'}). \tag{25}$$

where $Y(1, \mathcal{G}')$ represents the models output when segment $n$ is retained alongside the coalition $\mathcal{G}'$, and $Y(0, \mathcal{G}')$ represents the output when segment $n$ is masked, with only $\mathcal{G}'$ retained.

To establish the causal interpretability of SHAPEX at the model level, we reinterpret classical identification assumptions from causal inference in the space of model inputs [30, 31, 62]. Specifically:

- **Consistency & SUTVA**: Each shapelet-aligned segment is perturbed independently, and the effect of a given segment does not depend on the perturbation status of others. That is, the potential model output under a given intervention depends only on the subset of included segments, consistent with the SUTVA.

- **Ignorability**: Given a subset of retained segments $\mathcal{G}'$, the assignment variable $z \in \{0, 1\}$, representing whether segment $n$ is retained ($z = 1$) or masked ($z = 0$), is independent of the potential model outcomes. This holds because $z$ is controlled deterministically through our synthetic perturbation policy.

- **Positivity**: For any $\mathcal{G}'$, both interventions $z = 0$ and $z = 1$ occur with non-zero probability, ensured by our random sampling over subsets.

Under these assumptions, the model-level CATE $\tau_n^{\text{model}}$ is identifiable.

We now proceed to prove Proposition 1, which establishes the equivalence between the Shapley value computed by SHAPEX and the model-level CATE.

**Proposition 1** (Shapley Value as Model-Level CATE). *Let $f(X)$ denote the models prediction for input $X$, and let $X_{(S_n)}$ denote the segment aligned with shapelet $n \in \mathcal{G}$, where $\mathcal{G} = \{1, \ldots, N\}$ is the index set of all segments. Then, under the intervention defined by retaining or masking segment $X_{(S_n)}$, the Shapley value $\phi_n$ computed as*

$$\phi_n = \mathbb{E}_{\mathcal{G}'} \left[ f(X_{\mathcal{G}' \cup \{n\}}) - f(X_{\mathcal{G}'}) \right] \tag{26}$$

*is equivalent to the model-level CATE $\tau_n^{model}$, conditioned on context $\mathcal{G}' \subseteq \mathcal{G} \setminus \{n\}$.*

*Proof.* Given the validity of the three identification assumptions, the model-level CATE $\tau_n^{\text{model}}$ is identifiable. Its definition is given by:

$$\tau_n^{\text{model}} = \mathbb{E}_{\mathcal{G}'} \left[ Y(1, \mathcal{G}') - Y(0, \mathcal{G}') \right]. \tag{27}$$

Substituting the definitions of $Y(1, \mathcal{G}')$ and $Y(0, \mathcal{G}')$, we obtain:

$$\tau_n^{\text{model}} = \mathbb{E}_{\mathcal{G}'} \left[ f(X_{\mathcal{G}' \cup \{n\}}) - f(X_{\mathcal{G}'}) \right] = \phi_n \tag{28}$$

Thus, the Shapley value $\phi_n$ is equivalent to the model-level CATE $\tau_n^{\text{model}}$. $\qquad \square$

### D.2 Practical Estimation via Coalition Averaging

Using the temporally-relational subset strategy, we estimate:

$$\hat{\tau}_n^{\text{model}} = \frac{1}{M} \sum_{m=1}^{M} \left[ f(X_{\mathcal{G}_m \cup \{n\}}) - f(X_{\mathcal{G}_m}) \right],$$

where each $\mathcal{G}_m \subseteq \mathcal{G} \setminus \{n\}$ is sampled from segments temporally connected to $n$. This estimator is unbiased within the restricted coalition space, and reduces complexity from $O(|\mathcal{G}|!)$ to $O(|\mathcal{G}|)$.

**Approximation Caveat.** When only a subset of coalitions is used, the estimated CATE converges to a conditional average over a restricted support. We refer to this as a **shapelet-local CATE**, acknowledging that this is an approximation of the full-coalition effect.

Table 4: Overview of Datasets

| Category | Dataset Name | Classes | Testing Samples | Training Samples | Length of Series | Description | True Saliency Score |
|---|---|---|---|---|---|---|---|
| Real | UCR Archive (128 datasets) | varies | varies | varies | varies | Benchmark suite of diverse time series tasks | No |
| | ECG | 2 | 37,004 | 55,507 | 360 | Biomedical ECG signals with expert annotation | Yes |
| Synthesis | MCC-E | 2 | 2,000 | 10,000 | 800 | Shapelet count for classification, equal amplitude | Yes |
| | MCC-H | 2 | 2,000 | 10,000 | 800 | Shapelet count for classification, higher amplitude | Yes |
| | MTC-E | 2 | 2,000 | 10,000 | 800 | Shapelet type for classification, equal amplitude | Yes |
| | MTC-H | 2 | 2,000 | 10,000 | 800 | Shapelet type for classification, higher amplitude | Yes |

## E Experimental Details

### E.1 Dataset Details

Following the experimental setup of previous PHTSE methods [3, 4], we evaluate the explanation model using both synthetic and real-world datasets. The statistical details of the datasets are presented in Table 4.

**Synthetic Datasets.** Previous PHTSE methods [3, 4] have used synthetic dataset generation approaches such as SeqComb and FreqShapes, that derive from the data generation method in [63]. However Geirhos et al. [63] generates datasets by injecting highly salient features as shortcuts into the original data to test whether deep neural networks rely on them for classification. Accordingly, methods like FreqShapes inject time series segments with significantly higher amplitudes than the mean to serve as class-discriminative features. Therefore, this approach is more similar to synthetic data generation methods in time series anomaly detection, which primarily evaluate whether an explanation model can highlight timesteps with extreme amplitude variations. In real-world time series classification tasks, class-discriminative features are often defined by specific temporal patterns rather than simple amplitude differences. For example, in ECG data [8], the class-discriminative features P-waves and T-waves do not necessarily exhibit higher amplitudes than surrounding regions.

Based on this observation, we propose a new approach for synthetic data generation for the PHTSE problem. Inspired by the time series motif insertion methods in [37], we generate four different motif-based binary classification datasets: **Motif Count Classification Equal (MCC-E)**, **Motif Type Classification Large (MTC-L)**, **Motif Count Classification Large (MCC-L)**, and **Motif Type Classification Equal (MTC-E)**. Here, **"type"** and **"count"** refer to datasets where classification is determined by either the type of inserted motifs or their count, respectively. The terms **"Equal"** and **"Large"** indicate whether the inserted motif has the same amplitude as the baseline waveform or significantly exceeds it.

These dataset synthetic methods better align with the mechanisms of the generation real data, allowing us to better assess whether an explanation method can effectively identify time series patterns that are crucial for classification.

**Real-world Datasets.** To further evaluate the explanation methods, we conduct experiments on several real-world time series datasets across diverse domains. **ECG** [38] is a physiological dataset used for binary classification of atrial fibrillation (AF), where each instance corresponds to a raw ECG segment annotated by clinical experts. **UCR Archive** [39] is a large-scale benchmark suite containing time series from a wide range of application domains such as image outlines, motion sensors, and astronomical observations. Since most datasets in the archive do not provide ground-truth saliency labels, they are primarily used in our occlusion-based evaluation. Additionally, datasets with variable-length sequences are excluded for consistency. We select 114 fixed-length datasets from the archive that span a wide spectrum of domains and difficulty levels. *To the best of our knowledge, this is the first work in Post Hoc Time Series Explanation (PHTSE) that performs such a comprehensive and diverse evaluation on the UCR archive*.

## E.2 Benchmarks

We evaluate explanation methods using multiple benchmark approaches designed for interpreting deep learning models on time series data.

**Perturbation-based methods: TimeX++** [4] applies an information bottleneck principle to generate label-preserving, in-distribution explanations by perturbing the input while addressing trivial solutions and distributional shift issues. **TimeX** [3] trains an interpretable surrogate model to mimic a pretrained classifier while preserving latent space relations, ensuring faithful and structured explanations. **Dynamask** [2] generates dynamic perturbation masks to produce instance-wise importance scores while maintaining temporal dependencies. **WinIT** [23] captures feature importance over time by explicitly modeling temporal dependencies and summarizing importance over past time windows.

**Gradient-based methods: Integrated Gradients (IG)** [18] computes feature attributions by integrating gradients along a path from a baseline input to the actual input, ensuring sensitivity and implementation invariance. **SGT + GRAD** [19] enhances saliency-based explanations by reducing noisy gradients through iterative feature masking while preserving model performance. **MILLET** [40] enables inherent interpretability for time series classifiers by leveraging Multiple Instance Learning (MIL) to produce high-quality, sparse explanations without sacrificing performance.

**Metrics:** We follow the evaluation metrics from [3], adopting AUP (Area Under Perturbation Curve), AUR (Area Under Recall), and AUPRC (Area Under Precision-Recall Curve) in the saliency evaluation experiment to assess the discrepancy between the generated saliency scores and the ground truth. A metric value closer to 1 indicates higher accuracy. In the occlusion experiment, we use the prediction AUROC (Area Under the Receiver Operating Characteristic Curve) of the black-box classifier as the evaluation metric, measuring the impact of removing important subsequences on classification performance.

## E.3 Experimental Settings

All experiments were conducted on a machine equipped with an NVIDIA RTX 4090 GPU and 24 GB of RAM. The black-box classifiers used in our evaluation (e.g., Transformer, CNN) are trained independently using standard cross-entropy loss until convergence, with early stopping based on validation accuracy. Unless otherwise specified, default training settings from each baselines original implementation are followed.

Table 5: Saliency score evaluation on synthetic datasets (CNN as the black-box model).

| METHOD | MCC-E | | | MCC-H | | |
|---|---|---|---|---|---|---|
| | AUPRC | AUP | AUR | AUPRC | AUP | AUR |
| IG | 0.3394±0.0045 | 0.3966±0.0060 | 0.4689±0.0024 | 0.4097±0.0059 | 0.4896±0.0065 | 0.4960±0.0027 |
| DYNAMASK | 0.3488±0.0056 | 0.2277±0.0047 | 0.1008±0.0024 | 0.4017±0.0073 | 0.2633±0.0057 | 0.0510±0.0012 |
| WINIT | 0.1513±0.0028 | 0.0625±0.0010 | 0.4636±0.0074 | 0.1659±0.0033 | 0.0628±0.0011 | 0.4481±0.0063 |
| TIMEX | 0.2608±0.0029 | 0.2571±0.0032 | 0.5505±0.0013 | 0.3107±0.0033 | 0.3147±0.0038 | 0.4493±0.0013 |
| TIMEX++ | 0.3179±0.0036 | 0.3155±0.0041 | **0.5657**±0.0025 | 0.1960±0.0023 | 0.2109±0.0034 | 0.4900±0.0021 |
| SHAPEX | **0.6197**±0.0037 | **0.5157**±0.0078 | 0.2775±0.0048 | **0.8100**±0.0014 | **0.6144**±0.0045 | **0.7541**±0.0055 |

| METHOD | MTC-E | | | MTC-H | | |
|---|---|---|---|---|---|---|
| | AUPRC | AUP | AUR | AUPRC | AUP | AUR |
| IG | 0.3427±0.0017 | **0.7233**±0.0039 | 0.2616±0.0024 | 0.3531±0.0022 | 0.6482±0.0060 | 0.3664±0.0065 |
| DYNAMASK | 0.2869±0.0016 | 0.5252±0.0026 | 0.0857±0.0014 | 0.3300±0.0032 | 0.6177±0.0068 | 0.0798±0.0011 |
| WINIT | 0.1511±0.0034 | 0.0632±0.0018 | 0.4522±0.0068 | 0.1321±0.0021 | 0.0614±0.0008 | 0.4790±0.0066 |
| TIMEX | 0.1413±0.0015 | 0.1197±0.0012 | 0.5787±0.0022 | 0.2315±0.0032 | 0.2293±0.0040 | 0.5184±0.0027 |
| TIMEX++ | 0.1157±0.0009 | 0.0877±0.0008 | 0.4845±0.0035 | 0.1174±0.0010 | 0.1136±0.0014 | 0.5436±0.0026 |
| SHAPEX | **0.6954**±0.0029 | 0.5084±0.0051 | **0.6663**±0.0063 | **0.6793**±0.0014 | **0.4249**±0.0025 | **0.8933**±0.0045 |

To ensure robustness and account for variability in training and explanation outputs, we repeat each experiment across **five random seeds**, reporting the mean and standard deviation as error bars. The random seeds affect both model initialization and data shuffling. For methods involving sampling-based perturbation (e.g., TIMEX++), the same set of seeds is applied to ensure fair comparison.

All models and explanation methods are implemented in PyTorch, and our codebase supports efficient parallel evaluation across datasets and seeds.

For dataset splits, we follow the standard training/test partitions provided in each dataset. A validation set comprising **20% of the training set** is held out to tune hyperparameters and perform early stopping.

# F    Additional Experiments

## F.1    Saliency Experiments

To further examine the impact of black-box models on explainability, we conducted saliency evaluation experiments on both synthetic and real-world datasets using LSTM [34], CNN [35] and Multi-Rocket [36] as classifiers. The results are presented in Tables 5, 6, 8, 9, and 7.

We observe that other baseline methods are significantly affected by changes in the classifier, whereas SHAPEX consistently achieves the best performance with remarkable stability. This indicates that SHAPEX provides highly robust explanations, benefiting from its ability to preserve true causal relationships.

Table 6: Saliency score evaluation on ECG dataset (CNN as the black-box model).

| METHOD | ECG | | |
|---|---|---|---|
| | AUPRC | AUP | AUR |
| IG | 0.4949±0.0010 | 0.5374±0.0012 | 0.5306±0.0010 |
| DYNAMASK | 0.4598±0.0010 | 0.7216±0.0027 | 0.1314±0.0008 |
| WINIT | 0.3963±0.0011 | 0.3292±0.0020 | 0.3518±0.0012 |
| TIMEX | 0.6401±0.0010 | 0.7458±0.0011 | 0.4161±0.0008 |
| TIMEX++ | 0.6726±0.0010 | 0.7570±0.0011 | 0.4319±0.0012 |
| SHAPEX | **0.7198**±0.0029 | **0.8321**±0.0031 | **0.6948**±0.0032 |

Table 9: Saliency score evaluation on ECG dataset (LSTM as the black-box model).

| METHOD | ECG | | |
|---|---|---|---|
| | AUPRC | AUP | AUR |
| IG | 0.5037±0.0018 | 0.6129±0.0026 | 0.4026±0.0015 |
| DYNAMASK | 0.3730±0.0012 | 0.6299±0.0030 | 0.1102±0.0007 |
| WINIT | 0.3628±0.0013 | 0.3805±0.0022 | 0.4055±0.0009 |
| TIMEX | 0.6057±0.0018 | 0.6416±0.0024 | 0.4436±0.0017 |
| TIMEX++ | 0.6512±0.0011 | 0.7432±0.0011 | 0.4451±0.0008 |
| SHAPEX | **0.7206**±0.0028 | **0.8510**±0.0032 | **0.6924**±0.0032 |

Table 7: Saliency score evaluation on synthetic datasets (MultiRocket as the black-box model).

| | MCC-E | | | MCC-H | | |
|---|---|---|---|---|---|---|
| METHOD | AUPRC | AUP | AUR | AUPRC | AUP | AUR |
| IG | 0.2871±0.0065 | 0.4713±0.0013 | 0.3415±0.0086 | 0.3391±0.0048 | 0.4215±0.0038 | 0.5214±0.0069 |
| DYNAMASK | 0.4141±0.0048 | 0.4726±0.0058 | 0.4142±0.0014 | 0.4150±0.0079 | 0.1572±0.0038 | 0.2113±0.0033 |
| WINIT | 0.1974±0.0057 | 0.2411±0.0112 | 0.3250±0.0057 | 0.1528±0.0024 | 0.0925±0.0075 | 0.5101±0.0078 |
| CORTX | 0.3109±0.0071 | 0.4782±0.0171 | 0.4317±0.0148 | 0.5931±0.0016 | 0.5107±0.0011 | 0.4839±0.0062 |
| MILLET | 0.2355±0.0026 | 0.2513±0.0071 | 0.2142±0.0056 | 0.3147±0.0013 | 0.2511±0.0067 | 0.3125±0.0061 |
| TIMEX | 0.4032±0.0056 | 0.3140±0.0082 | 0.6490±0.0034 | 0.4513±0.0077 | 0.6712±0.0051 | 0.3500±0.0092 |
| TIMEX++ | 0.3509±0.0066 | 0.4851±0.0061 | 0.2631±0.0009 | 0.6351±0.0051 | 0.6524±0.0078 | 0.5120±0.0023 |
| SHAPEX_SF | 0.2715±0.0072 | 0.2231±0.0033 | 0.2690±0.0056 | 0.4681±0.0082 | 0.2046±0.0034 | 0.7135±0.0081 |
| SHAPEX | 0.6125±0.0067 | 0.5209±0.0023 | 0.3511±0.0083 | 0.8245±0.0044 | 0.6944±0.0070 | 0.7714±0.0023 |

| | MTC-E | | | MTC-H | | |
|---|---|---|---|---|---|---|
| METHOD | AUPRC | AUP | AUR | AUPRC | AUP | AUR |
| IG | 0.2481±0.0023 | 0.1125±0.0070 | 0.5721±0.0034 | 0.3152±0.0054 | 0.3513±0.0034 | 0.4742±0.0075 |
| DYNAMASK | 0.1409±0.0035 | 0.1266±0.0056 | 0.2519±0.0065 | 0.2760±0.0045 | 0.3937±0.0076 | 0.2175±0.0044 |
| WINIT | 0.1539±0.0076 | 0.1072±0.0023 | 0.0417±0.0050 | 0.1434±0.0076 | 0.0821±0.0023 | 0.4728±0.0078 |
| CORTX | 0.1841±0.0056 | 0.1851±0.0049 | 0.5152±0.0051 | 0.2153±0.0022 | 0.2307±0.0051 | 0.5311±0.0016 |
| MILLET | 0.1846±0.0027 | 0.1252±0.0009 | 0.1952±0.0024 | 0.2194±0.0038 | 0.2474±0.0089 | 0.3811±0.0032 |
| TIMEX | 0.2511±0.0061 | 0.6249±0.0010 | 0.1250±0.0043 | 0.3856±0.0008 | 0.3745±0.0039 | 0.1730±0.0023 |
| TIMEX++ | 0.2165±0.0042 | 0.4301±0.0097 | 0.2250±0.0042 | 0.3521±0.0061 | 0.3504±0.0057 | 0.1149±0.0055 |
| SHAPEX_SF | 0.3840±0.0042 | 0.1677±0.0021 | 0.4698±0.0003 | 0.3890±0.0041 | 0.4600±0.0041 | 0.1459±0.0082 |
| SHAPEX | 0.6705±0.0001 | 0.6558±0.0041 | 0.5660±0.0008 | 0.6859±0.0031 | 0.4262±0.0051 | 0.9122±0.0032 |

Table 8: Saliency score evaluation on synthetic datasets (LSTM as the black-box model).

| | MCC-E | | | MCC-H | | |
|---|---|---|---|---|---|---|
| METHOD | AUPRC | AUP | AUR | AUPRC | AUP | AUR |
| IG | 0.4752±0.0078 | 0.5282±0.0088 | 0.4622±0.0059 | 0.5144±0.0079 | 0.5297±0.0092 | 0.4965±0.0054 |
| DYNAMASK | 0.3999±0.0067 | 0.4925±0.0099 | 0.0369±0.0014 | 0.4178±0.0068 | 0.3239±0.0073 | 0.0103±0.0003 |
| WINIT | 0.2037±0.0076 | 0.1897±0.0160 | 0.3355±0.0085 | 0.2013±0.0057 | 0.1573±0.0157 | 0.3472±0.0115 |
| TIMEX | 0.2407±0.0026 | 0.2139±0.0025 | 0.5495±0.0009 | 0.5013±0.0052 | 0.5492±0.0065 | 0.3482±0.0019 |
| TIMEX++ | 0.3766±0.0041 | 0.3356±0.0040 | 0.5242±0.0017 | 0.4799±0.0043 | 0.2395±0.0039 | 0.5224±0.0046 |
| SHAPEX | 0.7422±0.0029 | 0.7507±0.0053 | 0.5703±0.0059 | 0.8166±0.0014 | 0.6893±0.0054 | 0.7701±0.0052 |

| | MTC-E | | | MTC-H | | |
|---|---|---|---|---|---|---|
| METHOD | AUPRC | AUP | AUR | AUPRC | AUP | AUR |
| IG | 0.2469±0.0031 | 0.3512±0.0063 | 0.4862±0.0044 | 0.4453±0.0029 | 0.5938±0.0030 | 0.3652±0.0023 |
| DYNAMASK | 0.1243±0.0012 | 0.0803±0.0038 | 0.0085±0.0005 | 0.2344±0.0016 | 0.2398±0.0058 | 0.0283±0.0006 |
| WINIT | 0.1395±0.0024 | 0.0953±0.0098 | 0.3785±0.0091 | 0.1560±0.0046 | 0.1457±0.0165 | 0.2519±0.0084 |
| TIMEX | 0.2012±0.0022 | 0.1500±0.0012 | 0.6780±0.0022 | 0.1874±0.0021 | 0.1556±0.0016 | 0.5216±0.0013 |
| TIMEX++ | 0.1314±0.0006 | 0.1235±0.0011 | 0.5070±0.0012 | 0.3169±0.0027 | 0.2076±0.0020 | 0.4632±0.0018 |
| SHAPEX | 0.6028±0.0053 | 0.4472±0.0080 | 0.5616±0.0086 | 0.6702±0.0015 | 0.3952±0.0024 | 0.8549±0.0056 |

## F.2 Details of the Occlusion Experiment

To enable a fair and comprehensive evaluation across diverse datasets, we first preprocessed the UCR Archive by excluding datasets with variable-length sequences, as their inconsistent input shapes are incompatible with fixed-length perturbation settings. Additionally, both TIMEX and TIMEX++ suffered from gradient explosion during training on a small subset of datasets. For fairness, we removed these failed runs from comparison.

Nevertheless, the remaining collection still covers a highly diverse and representative set of over 100 datasets, making it by far the most comprehensive occlusion-based evaluation conducted in the literature of PHTSE. The aggregated results are illustrated in Figure 8, 9, 10, and 11.

Given the inherent diversity of UCR datasets—covering various domains, sequence lengths, and class cardinalities, generating accurate saliency scores remains a challenging task. Consequently, no single explanation model dominates across all datasets. However, SHAPEX demonstrates superior robustness compared to baseline methods: its performance degrades more gracefully on challenging cases, as reflected in the smoother drop of AUROC scores across datasets. The statistical comparison, with Figure 6(a) reporting mean performance and Figure 6(b) highlighting the best-case results, reinforces the advantage of SHAPEX as the most consistently robust and reliable method for PHTSE tasks to date.

Table 10: Computation Time (in seconds) of Different Explanation Methods

| METHOD | MCC-E | MCC-H | MTC-E | MTC-H |
|---|---|---|---|---|
| TIMEX++ | 13.0470 | 13.7306 | 13.5581 | 5.6111 |
| IG | 49.7619 | 52.9600 | 53.4244 | 910.0483 |
| DYNAMASK | 837.2252 | 851.3321 | 840.9815 | 833.0267 |
| WINIT | 167.9867 | 165.1973 | 166.8685 | 166.5299 |
| SHAPEX | 129.0921 | 132.8452 | 127.1462 | 115.3432 |

Table 11: Ablation analysis on synthetic datasets.

| Ablations | MCC-E | | | MCC-H | | |
|---|---|---|---|---|---|---|
| | AUPRC | AUP | AUR | AUPRC | AUP | AUR |
| Full | **0.6407**±0.0036 | **0.5614**±0.0076 | 0.3679±0.0050 | **0.8113**±0.0013 | **0.6838**±0.0054 | **0.7431**±0.0055 |
| w/o matching loss | 0.1455±0.0112 | 0.1682±0.0057 | 0.1347±0.0012 | 0.2612±0.0035 | 0.1792±0.0011 | 0.3849±0.0023 |
| w/o diversity loss | 0.1309±0.0023 | 0.1450±0.0061 | 0.1103±0.0031 | 0.2630±0.0032 | 0.1849±0.0012 | 0.3542±0.0035 |
| w/o shapelet encoder | 0.2407±0.0013 | 0.0933±0.0032 | **0.4359**±0.0023 | 0.2366±0.0030 | 0.3274±0.0074 | 0.7113±0.0023 |
| w/o LINEAR | 0.5926±0.0036 | 0.5381±0.0011 | 0.3352±0.0021 | 0.7633±0.0024 | 0.6551±0.0062 | 0.7084±0.0037 |
| w/o segment | 0.6104±0.0036 | 0.5274±0.0090 | 0.2461±0.0046 | 0.7030±0.0023 | 0.6312±0.0099 | 0.3851±0.0002 |

| Ablations | MTC-E | | | MTC-H | | |
|---|---|---|---|---|---|---|
| | AUPRC | AUP | AUR | AUPRC | AUP | AUR |
| Full | **0.6100**±0.0048 | 0.3962±0.0067 | **0.5472**±0.0082 | **0.6792**±0.0014 | **0.4255**±0.0024 | **0.9019**±0.0041 |
| w/o matching loss | 0.1273±0.0052 | 0.1135±0.0057 | 0.1043±0.0056 | 0.2153±0.0034 | 0.1305±0.0044 | 0.3347±0.0012 |
| w/o diversity loss | 0.1371±0.0043 | 0.1524±0.0012 | 0.1042±0.0023 | 0.2094±0.0054 | 0.1053±0.0022 | 0.4914±0.0035 |
| w/o shapelet encoder | 0.2358±0.0057 | 0.1827±0.0076 | 0.0982±0.0023 | 0.1902±0.0030 | 0.2781±0.0012 | 0.8120±0.0033 |
| w/o LINEAR | 0.5627±0.0045 | **0.3991**±0.0033 | 0.5184±0.0055 | 0.6204±0.0065 | 0.3683±0.0042 | 0.7711±0.0055 |
| w/o segment | 0.5792±0.0070 | 0.3518±0.0045 | 0.5371±0.0020 | 0.6347±0.0045 | 0.4014±0.0060 | 0.8820±0.0034 |

# G  Further Analysis Experiments

## G.1  Computational Cost Analysis

It is well known that computing Shapley value is highly time-consuming, making computational cost a critical consideration in Shapley value analysis. We measure the inference time required for explanation generation across several datasets, with results presented in Table 10.

While SHAPEX is not the fastest method, it is significantly more efficient than Dynamask. This acceptable computational cost is primarily attributed to SHAPEX's segment-level design and its temporally relational subset extraction strategy. Consequently, SHAPEX not only achieves superior performance but also maintains practical feasibility.

## G.2  Parameter Analysis

We further analyze the impact of shapelet-related parameters on the performance of SHAPEX. Specifically, we examine the interplay between shapelet length ($L$) and the number of shapelets ($N$). Figure 12 presents the results on the ECG dataset.

Overall, we observe that AUP decreases as the number of shapelets increases, indicating that only shapelets corresponding to key features contribute to improved accuracy. In contrast, AUR exhibits an opposite trend, suggesting that a larger number of shapelets facilitates the discovery of more latent features. Additionally, optimal performance is achieved when the shapelet length is appropriately chosen, highlighting the importance of selecting a balanced length.

## G.3  Case Study

To qualitatively assess the interpretability of SHAPEX, we present case studies on three representative UCR datasets with well-defined domain semantics: PhalangesOutlinesCorrect, FaceAll, and UWaveGestureLibraryAll. These datasets span the domains of medical imaging, biometric contours, and motion sensors, respectively. In each case, we compare SHAPEX against existing baselines by visualizing the generated saliency scores.

**PhalangesOutlinesCorrect Dataset.** The PhalangesOutlinesCorrect dataset [64] is a one-dimensional time series derived from X-rays of the third phalanx through contour-based processing. First, the bone contour is extracted from the original radiograph. Then, the Euclidean distance from

each point on the contour to the bones center is computed. By radially sampling these distances in a clockwise direction, a polar distance sequence is formed, which captures the morphological profile of the bone outline in 1D space.

As illustrated in Figure 13, the sequence may contain two components: the shaft and the epiphysis. If a growth plate is present, the sequence includes a distinct transition region—highlighted in greenbetween the bone shaft and the epiphysis. The dataset comprises two classes: *Immature*, corresponding to the early Tanner-Whitehouse (TW) stages, and *Mature*, representing the later stages. In clinical practice, the key criterion for distinguishing between these two classes is the extent of epiphyseal development, particularly its fusion with the diaphysis. In the time series representation, the growth plate transition region typically corresponds to the time interval between steps 40 and 60.

In Figure 14, we observe that, as the most advanced baselines, TIMEX and TIMEX++ predominantly generate explanations focused on the peaks or valleys of the sequence. However, these features are not the primary indicators of bone maturity. In contrast, SHAPEX highlights the sequence region between the 40th and 60th time steps, which is critical for reflecting the developmental status of the epiphysis. This indicates that SHAPEX effectively identifies segments within the sequence that are medically significant, distinguishing it significantly from other baselines that focus on prominent values in the sequence.

**FaceAll Dataset.** The FaceAll dataset [39] is sourced from the UCR archive. It consists of head contours collected from 14 graduate students, converted into one-dimensional time series through a series of image processing algorithms. The first half of the sequence corresponds to the facial contour of the human head, while the second half corresponds to the contour of the hair. This dataset includes 14 classes, each representing one of the 14 graduate students. Due to the high variability in human hair shapes, the classification results are primarily determined by the facial contour portion of the time series.

In the visualization of model predictions, as shown in Figure 15, we observe that only SHAPEX effectively highlights the facial contour regions, whereas other models merely focus on a few peaks and fail to provide meaningful insights.

**UWaveGestureLibraryAll Dataset.** The UWaveGestureLibraryAll dataset [41] generates sequences by recording acceleration signals during user gestures, encompassing eight predefined gestures. Compared to the phases of increasing acceleration, the phases where acceleration begins to decrease often carry more gesture-specific information. This is because decreasing acceleration indicates the beginnings of gesture transitions, such as sharp turns or circular motions.

In Figure 16, we observe that SHAPEX distinctly focuses on regions with negative slopes in the sequence, which correspond to the phases where gestures begin to change. This demonstrates that our method can effectively identify sub-sequences that are more valuable for sequence classification.

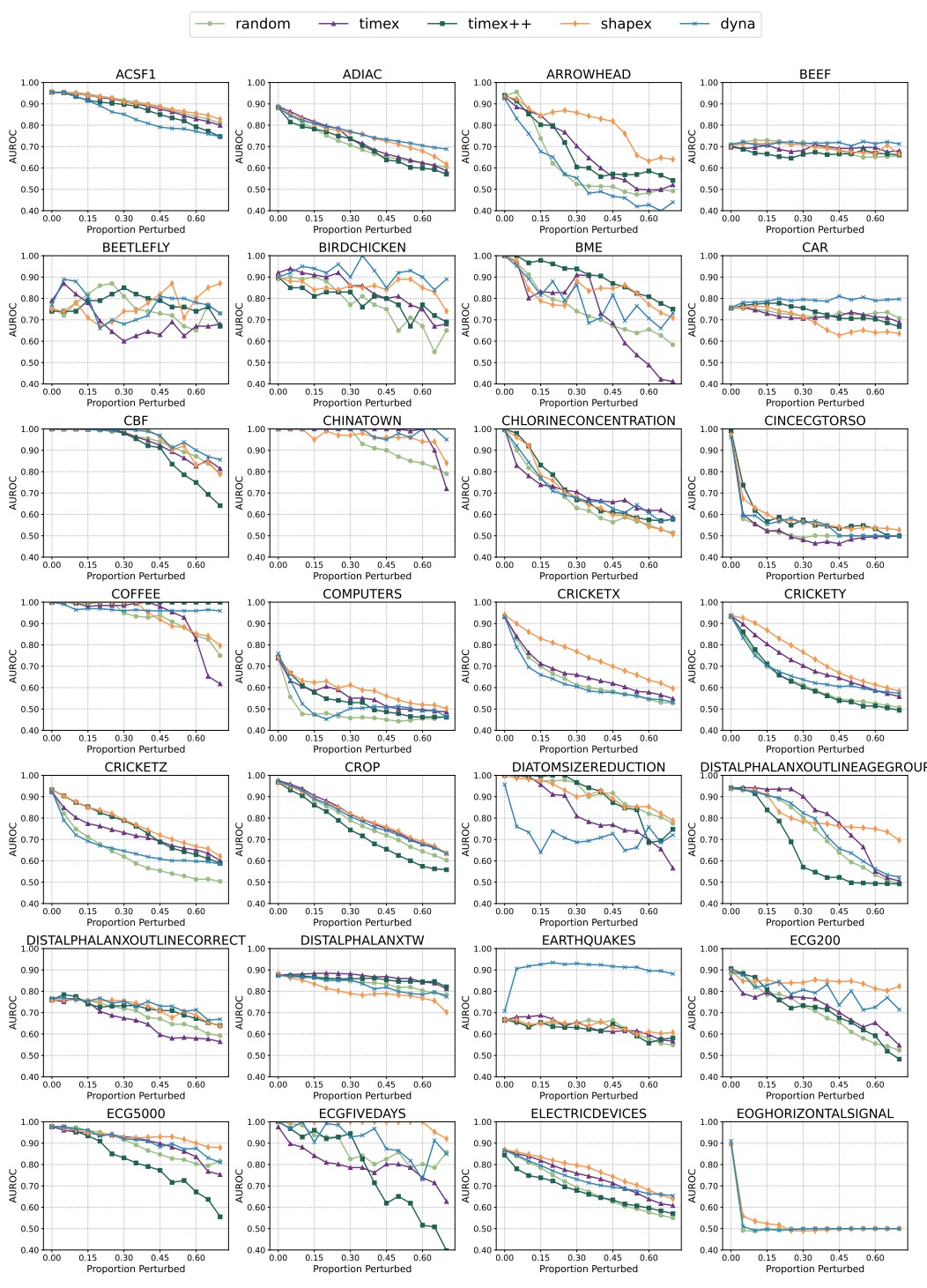

Figure 8: Occlusion experimental results on UCR archive.

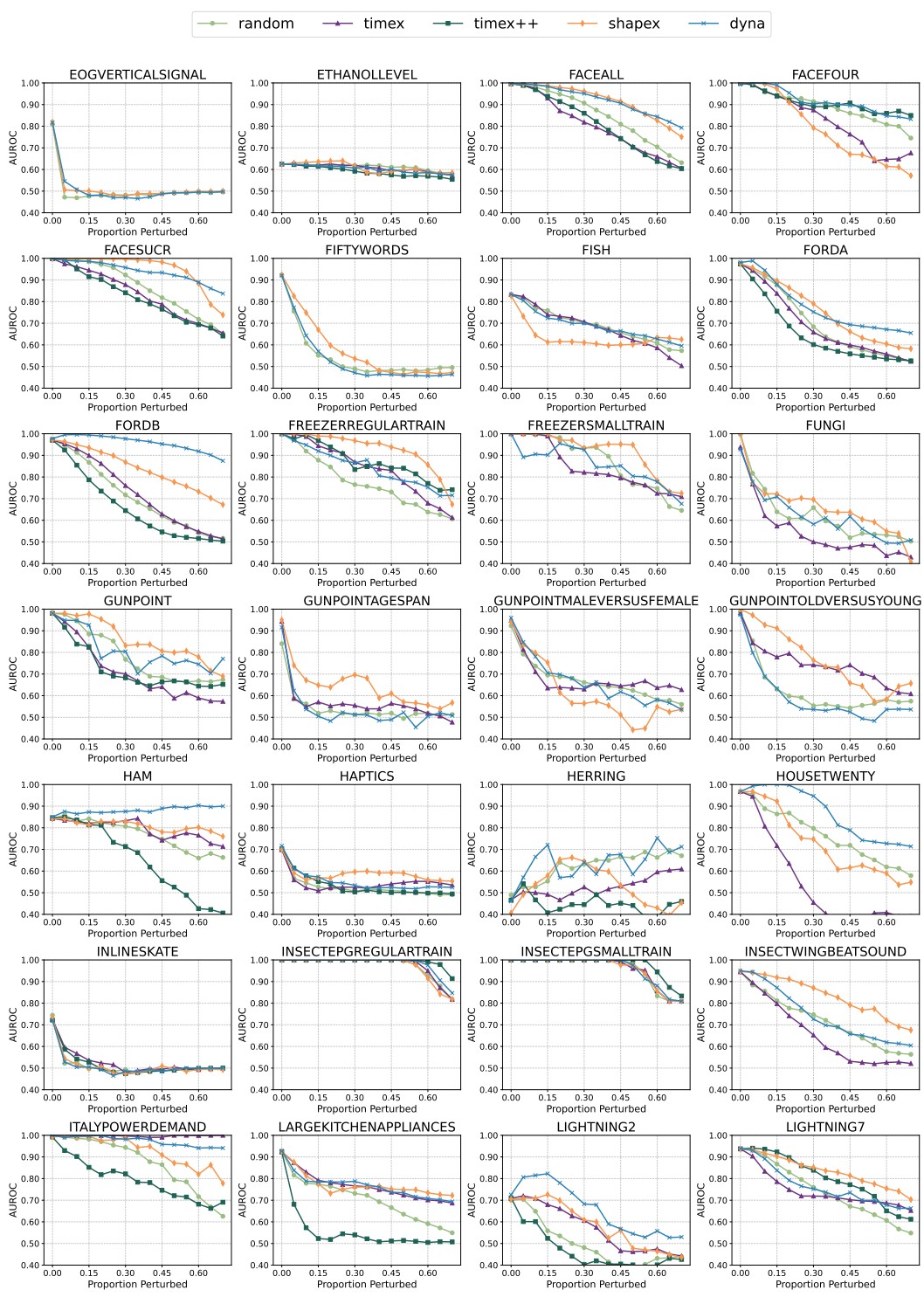

Figure 9: Occlusion experimental results on UCR archive.

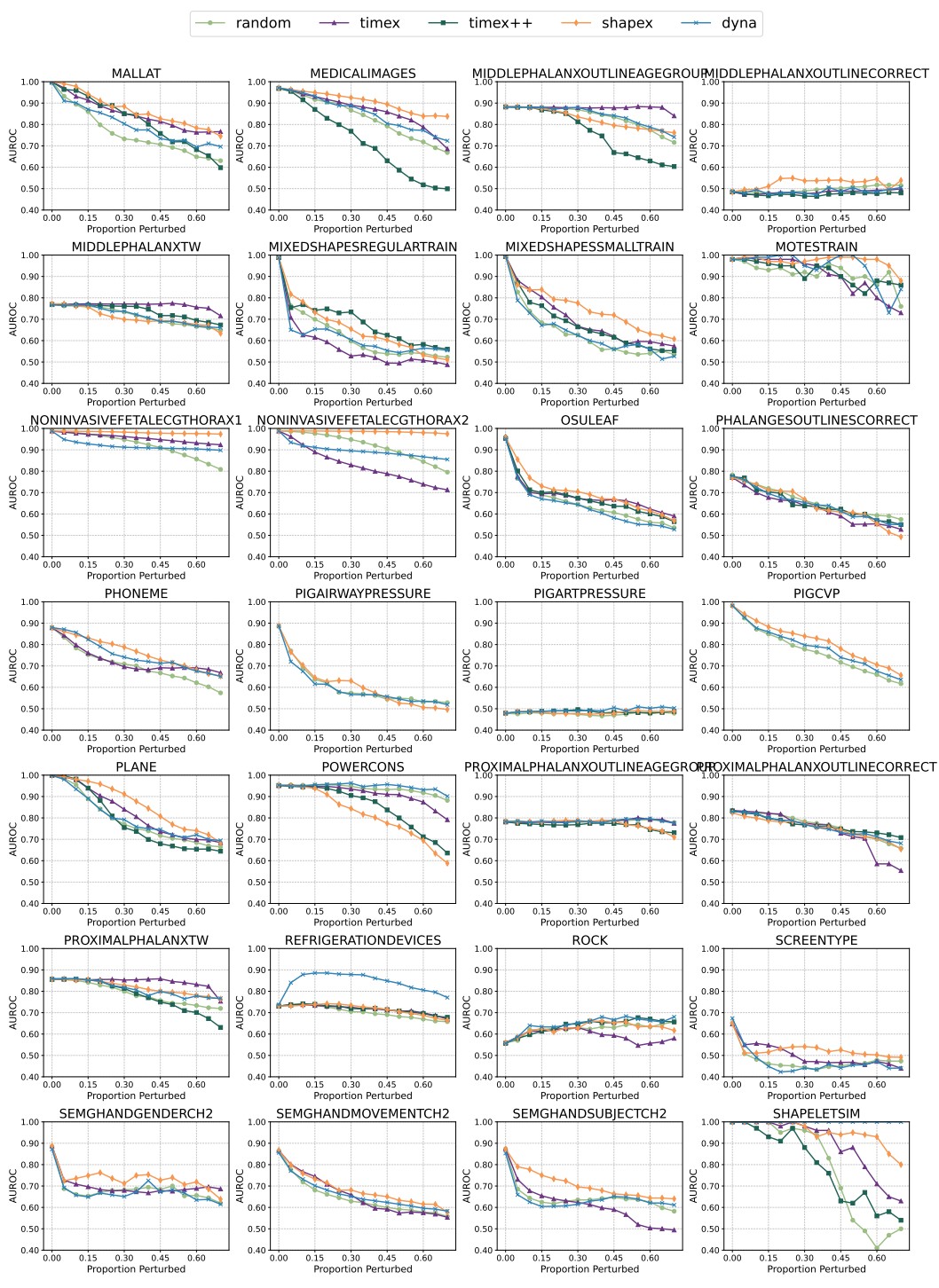

Figure 10: Occlusion experimental results on UCR archive.

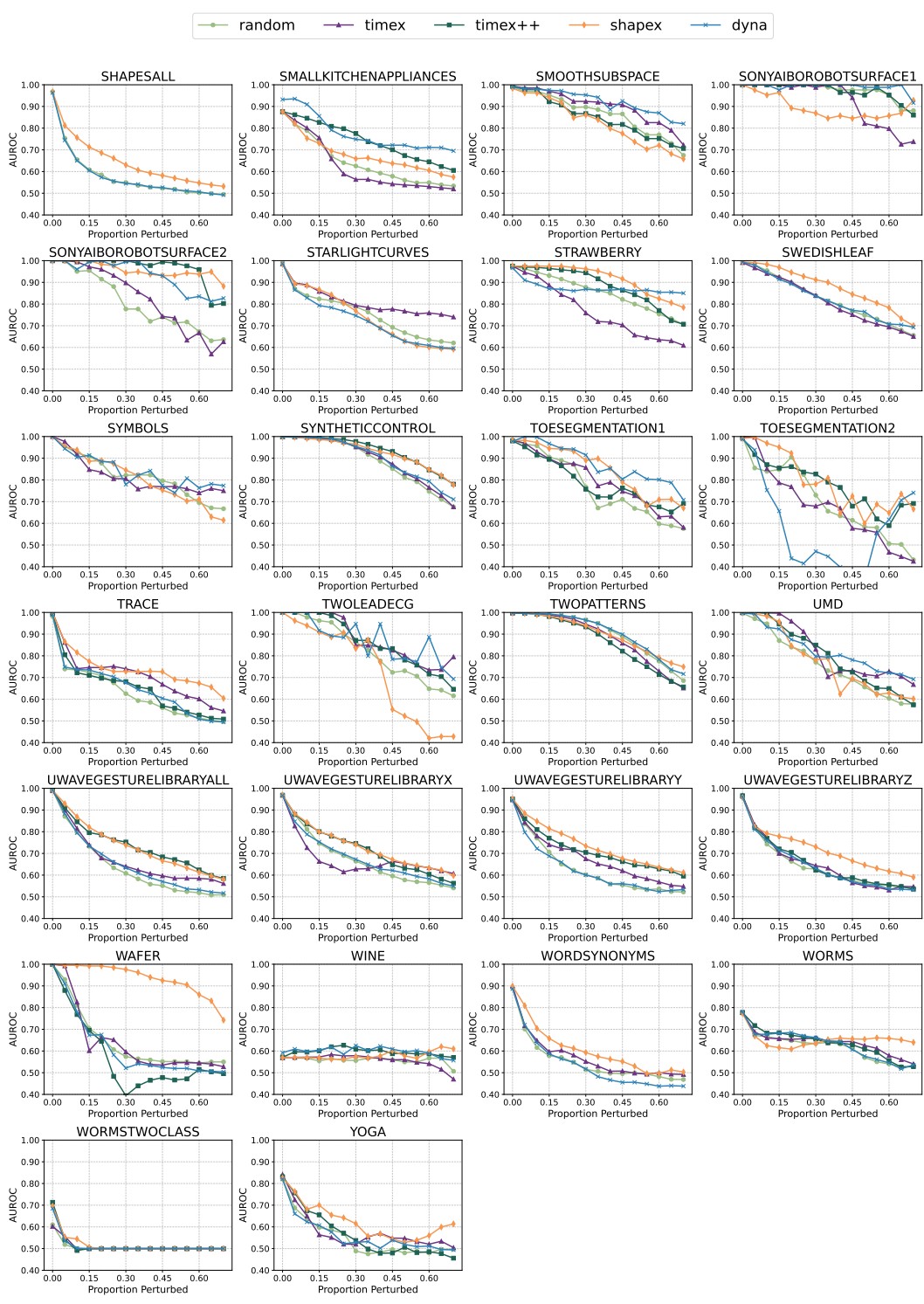

Figure 11: Occlusion experimental results on UCR archive.

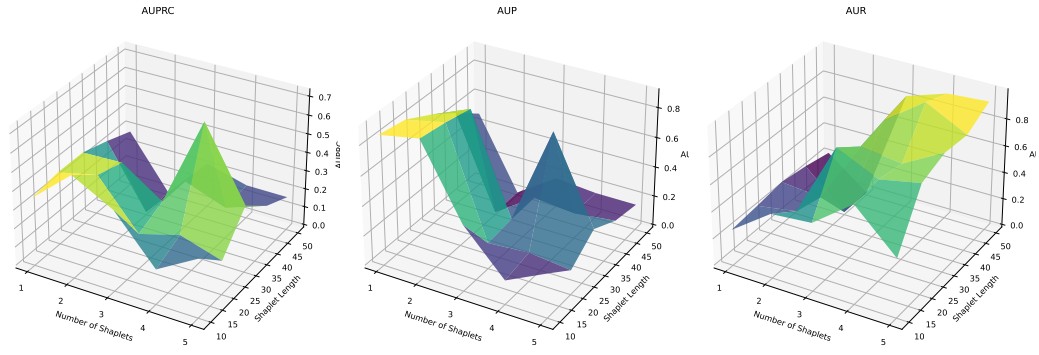

Figure 12: Saliency score generated by SHAPEX concerning various shapelet lengths and quantities on ECG Dataset.

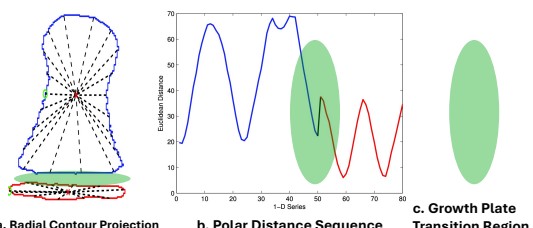

**a. Radial Contour Projection**    **b. Polar Distance Sequence**    **c. Growth Plate Transition Region**

Figure 13: Radial sampling from the center (a) yields a polar distance sequence (b), where the green region highlights the growth plate transition region. [64]

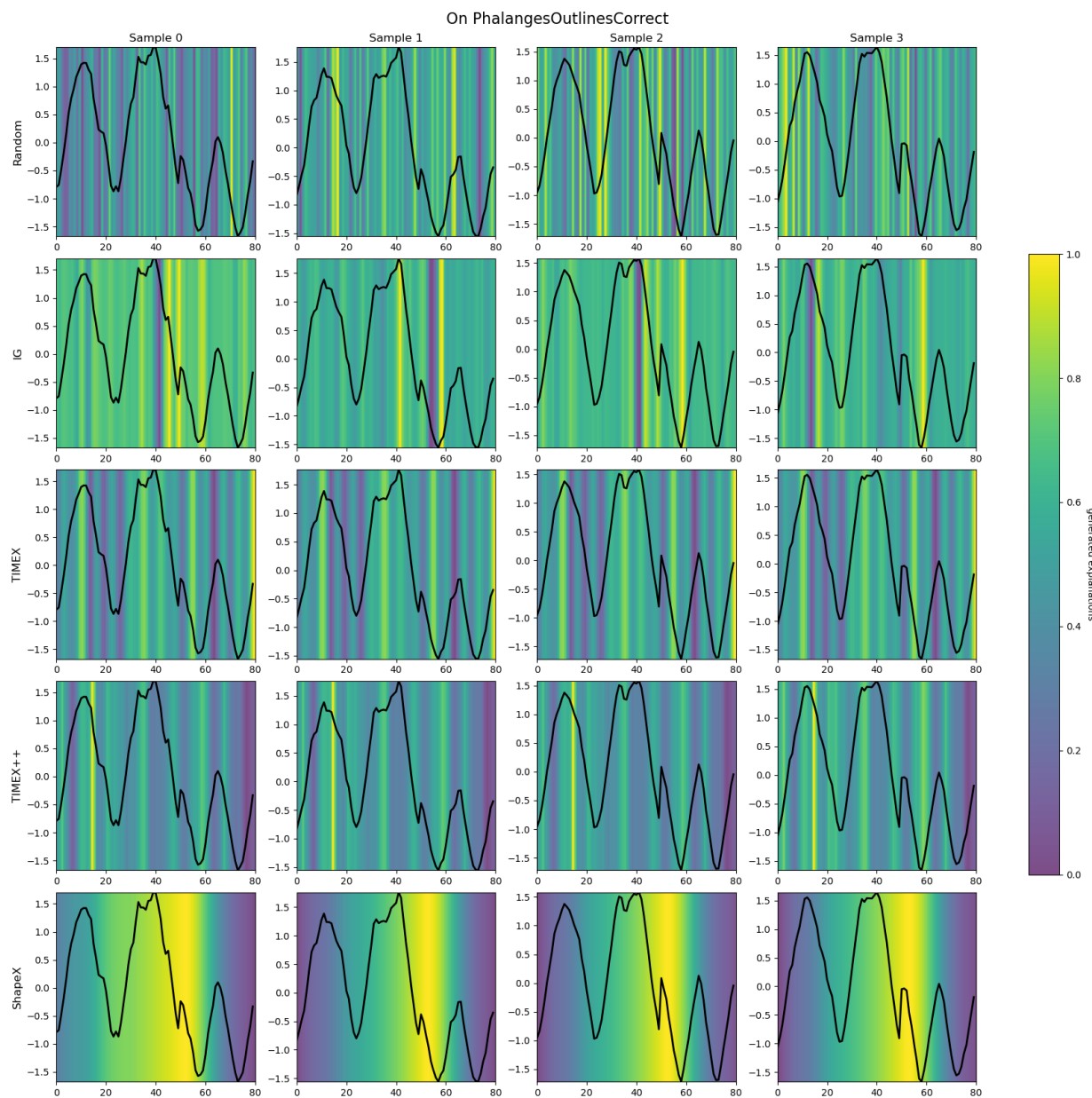

Figure 14: Visualization of saliency score on PhalangesOutlinesCorrect dataset.

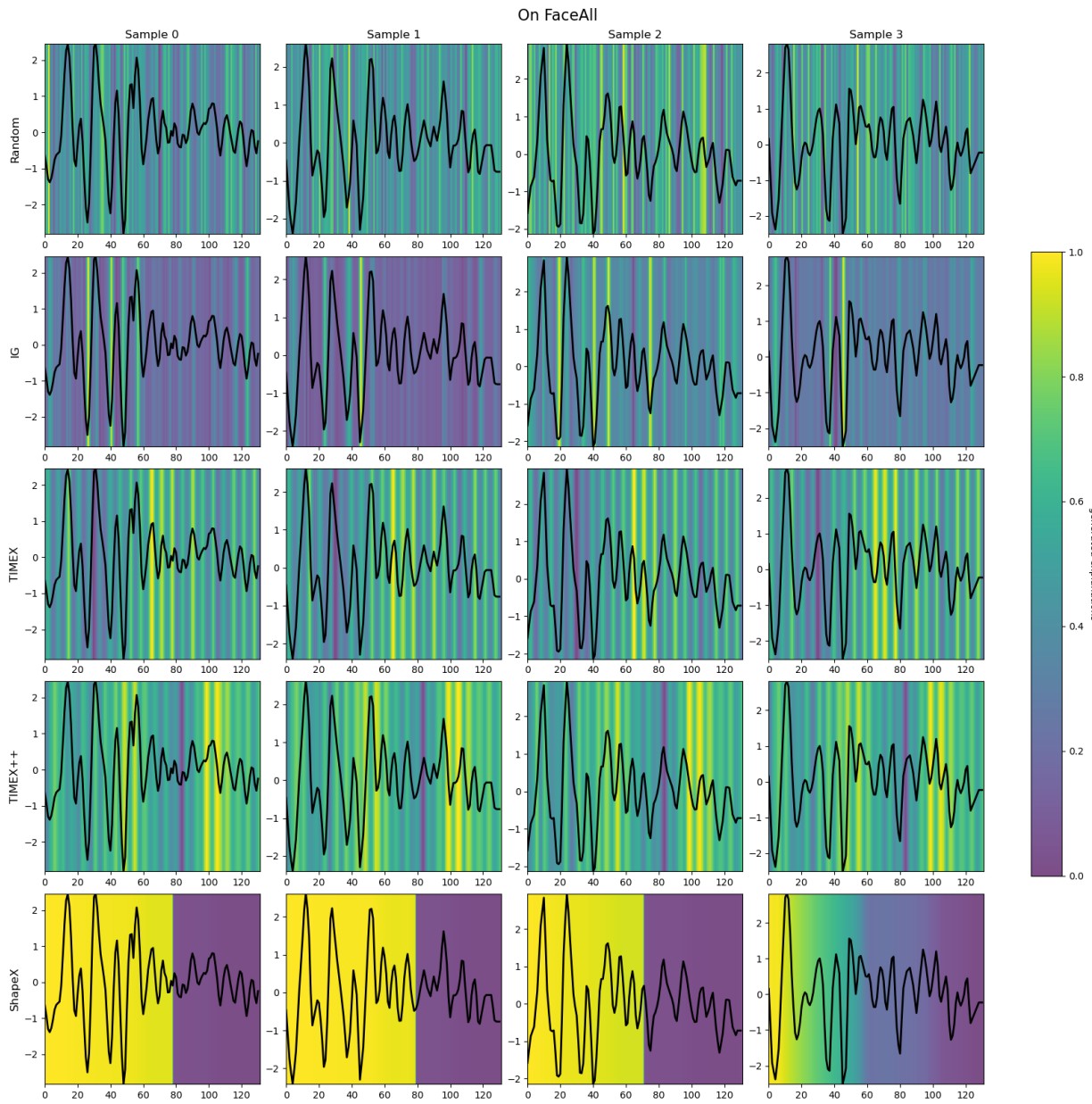

Figure 15: Visualization of saliency score on FaceAll dataset.

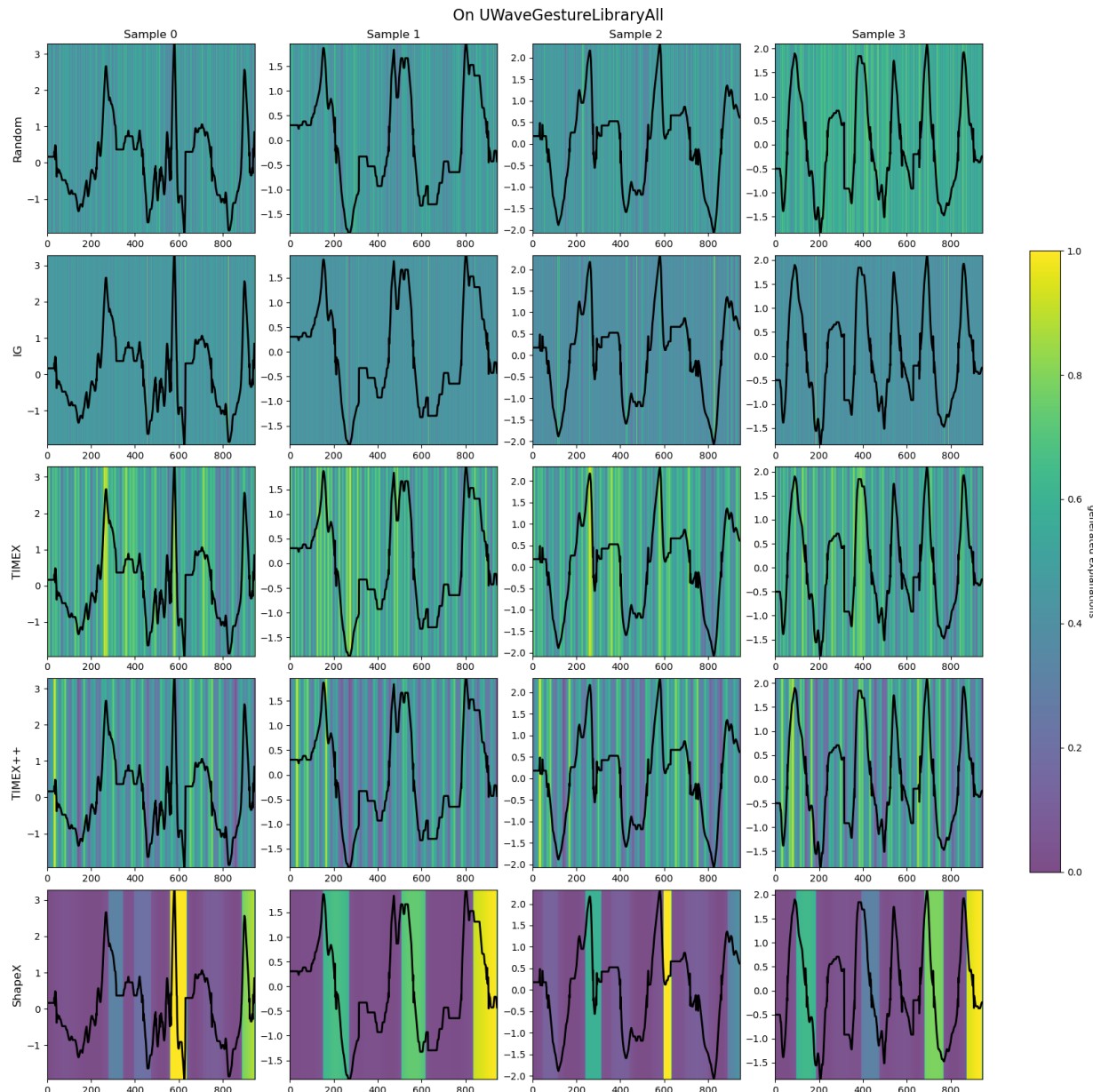

Figure 16: Visualization of saliency score on UWaveGestureLibraryAl dataset.

