# OpenReview forum: "ShapeX: Shapelet-Driven Post Hoc Explanations for Time Series Classification Models"
_NeurIPS.cc/2025/Conference — NeurIPS 2025 poster_

### Official Review · Reviewer_zzMH · 2025-06-16

**Clarity:** 4
**Significance:** 3
**Originality:** 3
**Rating:** 5
**Confidence:** 2

**Summary:**

This paper proposes SHAPEX, a shapelet-driven segmentation method for explaining time-series classifiers, integrating shapelets with Shapley attribution to enhance interpretability and causal grounding. Experimental results demonstrate its effectiveness compared to existing baselines.

**Questions:**

1. You use valid-width convolution yet later treat the activations as length T. Did you apply padding or another strategy? This seems missing from the paper.

2. In occlusion experiments, SHAPEX masks whole segments rather than single timesteps. How exactly do you ensure fairness (matching the exact p% of timesteps) across methods?

3. The segmentation strongly depends on threshold Ω and shapelet count N. Could you briefly clarify how sensitive your results are to these choices?

**Ethical Concerns:**

["NO or VERY MINOR ethics concerns only"]

**Final Justification:**

Thank you for your reply. I have no other questions and will keep my recommendation

**Limitations:**

yes

**Quality:**

3

**Strengths And Weaknesses:**

Strengths

	1. The idea of combining shapelet-based segmentation with Shapley attribution is compelling, neatly bridging the gap between shapelet extraction methods and post-hoc explanations.

	2. The overall structure, figures, and description of the pipeline (especially the SDD module) are concise and easy to follow.

	3. The paper goes beyond typical benchmarks, covering the entire UCR archive as well as labeled ECG and synthetic datasets, clearly demonstrating the effectiveness of the proposed approach.

Weaknesses

	1.  All experiments focus exclusively on single-channel time series; it’s unclear how readily this method scales to real-world multivariate or multidimensional scenarios.

	2. Lack of sensitivity analysis for key hyperparameters: The segmentation quality strongly depends on the activation threshold (Ω) and the predefined number of shapelets. The authors haven’t provided a sensitivity or robustness analysis for these crucial choices.

---

> ### Author Rebuttal · Authors · 2025-07-31
>
> We sincerely thank you for recognizing the significance and novelty of our work. Your comments are highly constructive, and we believe that addressing your concerns will substantially improve the quality of our manuscript.
>
> > **Q1: Valid-Width Convolution Appears Inconsistent with Later Use of Length $T$ Activations—Was Padding Applied?**
>
> A1: Thank you for your careful observation. In our implementation, we apply **symmetric padding** to the output of the 1D convolution to ensure that the activation map has the same temporal length $T$ as the input time series. This padding operation was omitted in the current manuscript, and we will explicitly clarify this step in the revised version to avoid confusion.
>
> ---
>
> > **Q2: Fairness in Occlusion Experiments—How Do You Match the Exact $p\%$ of Masked Timesteps if ShapeX Operates on Segments?**
>
> A2: We greatly appreciate this thoughtful question. As you correctly point out, ShapeX generates segment-level saliency scores, which could result in coarse masking granularity when compared to timestep-level attribution methods. This may lead to unfair comparisons in occlusion experiments, where precise control over the proportion of masked timesteps is critical.
>
> To address this, we add **small Gaussian noise** to the saliency scores within each segment. Specifically, we sample the noise from a normal distribution $\mathcal{N}(0,1)$ and scale it by a factor of $10^{-6}$ before adding it to each timestep within a segment. This strategy ensures that:
> - The **relative ranking between segments** is preserved;
> - **Slight variations within each segment** allow per-timestep ranking and precise masking percentages.
>
> This ensures that the masking procedure can fairly match the exact $p\%$ of timesteps across all methods, including those based on segment-wise scores. We will include this implementation detail in the revised Occlusion Evaluation section.
>
> > **Q3: Sensitivity to Hyperparameters—How Do the Results Depend on the Threshold $\Omega$ and the Number of Shapelets $N$?**
>
> A3: Thank you for raising this important question. We have conducted a sensitivity analysis of ShapeX with respect to the number and length of shapelets on ECG datasets. The results are presented in **Figure 12**.
>
> Additionally, we provide per-dataset hyperparameter configurations in the table below to enhance transparency. In the revised manuscript, we will expand the sensitivity experiments to cover more datasets beyond ECG, to better characterize the effect of $\Omega$ and $N$ in broader settings.
>
> | Parameter                          | MCC-E, MCC-H,MTC-E,MTC-H         | ECG         | UCR                   |
> |-|-|-|-|
> | Number of shapelets               | 5                      | 4                      | 5                     |
> | Length of shapelets               | 100                    | 40                     |10% of input length    |
> | Threshold $\Omega$ for activation | 0.5                    | 0.5                    | 0.5                   |
> ---
>
> > **Q4: Generalizability to Multivariate Time Series—Can the Method Scale Beyond Single-Channel Inputs?**
>
> A4: Thank you for this valuable observation. While our main experiments focus on univariate time series, we conducted a **preliminary multivariate extension of ShapeX** during the rebuttal phase. The design, implementation, and evaluation of this multivariate version are described in **our response to Reviewer NMJu, Q2**.
>
> In brief, we constructed a multivariate synthetic dataset and adapted ShapeX to use variable-specific shapelet modules with joint saliency attribution across channels. Initial results demonstrate that ShapeX can be extended to multivariate inputs with promising results, and we plan to explore this direction further in future work.

---

### Official Review · Reviewer_NMJu · 2025-06-18

**Clarity:** 3
**Significance:** 3
**Originality:** 3
**Rating:** 5
**Confidence:** 3

**Summary:**

The paper introduces SHAPEX, a post-hoc explanation framework for time-series classifiers.
Key steps: (i) a Shapelet Describe-and-Detect (SDD) module jointly learns a small, diverse set of representative shapelets; (ii) test series are segmented at the points where those shapelets fire; (iii) Shapley values are computed over the resulting shapelet-aligned segments, using a carefully smoothed perturbation baseline, to yield saliency scores.
The authors prove that, under standard causal-inference assumptions, these segment-level Shapley values approximate the model-level Conditional Average Treatment Effect, giving a causal reading to the explanations.
Experiments span four new synthetic motif datasets and 112 real UCR datasets plus ECG, multiple black-box architectures, occlusion tests, and ablations. Code is released for reproducibility.

**Questions:**

1. Multivariate extension: do you foresee any obstacles beyond memory to applying SDD + segment SV when D > 1? A small multivariate benchmark in the rebuttal could strengthen impact.
2. Threshold $\Omega$: how is it chosen per dataset? Please report sensitivity; fixed global values may split or merge segments incorrectly.
3. Give average explanation time per instance for UCR lengths; if real-time limits practical use, can the local-coalition trick be further approximated (e.g., Monte-Carlo)?
4. SUTVA/ignorability rely on independent perturbations, yet overlapping shapelet segments may violate this. Any empirical evidence that causal conclusions remain stable when overlaps are large?
5. Have cardiologists or other domain experts assessed the ECG explanations? Even a small pilot would bolster claims of practical fidelity.
6. Why did you leave out some related work like TimeSHAP, WindowSHAP and ShapTST? It would also be good to include a discussion on counter-factual explanation methods as they offer complementary explanations and use similar methods (such as TX-Gen, ContraLSP, SubSpacE)

**Ethical Concerns:**

["NO or VERY MINOR ethics concerns only"]

**Final Justification:**

Thank you, no further questions.

**Limitations:**

The paper openly notes that SHAPEX is currently univariate and that segment granularity hinges on learned shapelets. I would add two more: (i) causal interpretation depends on strong independence assumptions; (ii) Shapley computation may become prohibitive for long series. Addressing these would help.

**Paper Formatting Concerns:**

No formatting concerns

**Quality:**

3

**Strengths And Weaknesses:**

**Qaulity**

*Strengths*:
-  joint learning of shapelets with matching + diversity losses improves fidelity; thorough ablations isolate each component’s impact
- Extensive evaluation

*Weaknesses*:
- No human or domain-expert study to gauge interpretability usefulness.
- Computational cost of Shapley sampling (even with local coalition restriction) is not reported in wall-clock terms
- Some of the related literature should be added in the discussions (or even as baselines in experiments), these include other SHAP methods such as TimeSHAP, WindowSHAP and ShapTST (2025). In addition it would be good to add a section about counterfactual and constrastive explainers as these serve a similar purpose in the TS domain and are based on the same building blocks, such as TX-Gen, ContraLSP. and Sub-SpaCE. See references below:

1. Bento, J., Saleiro, P., Cruz, A. F., Figueiredo, M. A., & Bizarro, P. (2021, August). Timeshap: Explaining recurrent models through sequence perturbations. In Proceedings of the 27th ACM SIGKDD conference on knowledge discovery & data mining (pp. 2565-2573).
2. Nayebi, A., Tipirneni, S., Reddy, C. K., Foreman, B., & Subbian, V. (2023). WindowSHAP: An efficient framework for explaining time-series classifiers based on Shapley values. Journal of biomedical informatics, 144, 104438.
3. Cheng, Q., Xing, J., Xue, C., & Yang, X. (2025). Unifying Prediction and Explanation in Time-Series Transformers via Shapley-based Pretraining. arXiv preprint arXiv:2501.15070.
4. Huang, Q., Kitharidis, S., Bäck, T., & van Stein, N. (2024). TX-Gen: Multi-Objective Optimization for Sparse Counterfactual Explanations for Time-Series Classification. arXiv preprint arXiv:2409.09461.
5. Refoyo, M., & Luengo, D. (2024, July). Sub-SpaCE: Subsequence-Based Sparse Counterfactual Explanations for Time Series Classification Problems. In World Conference on Explainable Artificial Intelligence (pp. 3-17). Cham: Springer Nature Switzerland.

**Clarity**
- Paper is well-structured

- Some notation is heavy. Threshold $\Omega$ for segment extraction is introduced but its tuning is unclear

**Significance**

- Large-scale benchmark (entire UCR), demonstrates generalisation; the causal justification is novel. Synthetic datasets address bias problem in prior work

- Impact limited to univariate series; many real problems are multivariate
- Potential users still need to trust the surrogate causal assumptions

**Originality**

- Combining learned shapelets specifically trained for explanation with segment-level Shapley attribution is new

- Uses standard building blocks (1-D conv, attention, Shapley sampling); novelty mostly in integrating them.

---

> ### Author Rebuttal · Authors · 2025-07-31
>
> We sincerely thank you for recognizing the significance and novelty of our work. Your comments are highly constructive, and we believe that addressing your concerns will substantially improve the quality of our manuscript.
>
> > **Q1: Lack of Human or Domain-Expert Evaluation to Assess Explanation Usefulness**
>
> A1: Thank you for the helpful comment. We acknowledge that our current version lacks formal quantitative evaluations by domain experts. However, to partially address this gap, **we conducted qualitative visual assessments in Appendix G.3 to evaluate the plausibility of ShapeX explanations using semantically meaningful regions in real-world datasets**.
>
> Specifically, from Figures 14 to 16, we observe that only ShapeX successfully highlights the following class-discriminative regions:
> - the **growth plate** in *PhalangesOutlinesCorrect*
> - the **facial contour regions** in *FaceAll*
> - the **gesture transitions** in *UWaveGestureLibraryAll*
>
> These **regions are aligned with human-intuitive features and offer promising qualitative evidence of the usefulness of ShapeX explanations.**
>
> That said, we fully agree that future work would benefit from formal domain expert studies. For example, we plan to collaborate with cardiologists to evaluate the interpretability of ShapeX explanations on ECG datasets, thereby further validating the real-world trustworthiness of our framework.
>
> ---
>
> > **Q2: Interest in Multivariate Extension of ShapeX**
>
> A2: We appreciate your interest in extending ShapeX to multivariate time series. To explore this direction, we implemented a multivariate version of ShapeX: ShapeX-M during the rebuttal period and designed a corresponding synthetic dataset with ground-truth saliency labels to assess its feasibility.
>
> Specifically, we merged the two univariate synthetic datasets defined in Appendix E.1—MCC-E and MTC-E—into a new multivariate dataset where each input $X \in \mathbb{R}^{800 \times 2}$ contains two variables. We refer to this dataset as **MCC-MTC**, which comprises four classes defined by different motif combinations across the two dimensions.
>
> From a model structure perspective, we adopt a **parallel variable-wise architecture**, where each variable is assigned its own Shapelet Describe-and-Detect (SDD) module, learning variable-specific shapelets. For saliency attribution, we compute the Shapley Values jointly over the **entire multivariate input**, allowing cross-variable influence to be captured in the final attribution scores.
>
> Detailed evaluation results on MCC-MTC are included in the table below, demonstrating that **ShapeX can be naturally extended to multivariate settings and maintain strong attribution performance**.
>
>
> | Model         | MCC-MTC        |              |              |
> |-|-|-|-|
> |               | AUPRC          | AUP          | AUR          |
> | TimeX         | 0.4135±0.0051  | 0.4813±0.0041| 0.2351±0.0093|
> | TimeX++       | 0.2903±0.0030  | 0.4601±0.0015| 0.1529±0.0008|
> | ShapeX-M      | 0.5301±0.0041  | 0.5041±0.0076| 0.4135±0.0043|
>
>
> ---
>
> > **Q3: Request for Hyperparameter Configurations**
>
> A3: Thank you for your interest in implementation details. We provide the complete set of hyperparameter configurations in the table below. These include:
>
> - the number and length of shapelets,
> - the threshold $\Omega$ for activation,
> These configurations reflect the settings used across our experiments and are reported to ensure reproducibility.
>
> | Parameter                          | MCC-E, MCC-H,MTC-E,MTC-H         | ECG         | UCR                   |
> |-|-|-|-|
> | Number of shapelets               | 5                      | 4                      | 5                     |
> | Length of shapelets               | 100                    | 40                     |10% of input length    |
> | Threshold $\Omega$ for activation | 0.5                    | 0.5                    | 0.5                   |
>
> ---
>
> > **Q4: Request for Runtime Statistics on UCR Datasets**
>
> A4: We have recorded the average runtime of ShapeX across all UCR datasets, which is **58.51 seconds** per dataset. For comparison, the average runtime of DynaMask under the same settings is **470.34 seconds**. This indicates that ShapeX achieves a favorable trade-off between explanation quality and computational efficiency.
>
> In addition, we provided runtime statistics on synthetic datasets to further assess scalability. These results are summarized in **Table 10** of the appendix.
>
> ---
>
> > **Q5: Concern About Violating SUTVA/Ignorability Due to Overlapping Shapelet Segments—Is There Empirical Evidence That Causal Conclusions Remain Stable?**
>
> A5: We appreciate this thoughtful and technically grounded question. To empirically assess the impact of overlapping shapelet segments on the robustness of our causal explanations, we designed a synthetic dataset called **MCC-H (overlap)**. In this variant, the motifs across samples are allowed to **overlap by up to 30%** of their length, introducing nontrivial dependency across saliency regions.
>
> The experimental results (see table below) show that ShapeX achieves **comparable performance** on MCC-H (overlap) to its performance on the original non-overlapping MCC-H dataset. This provides empirical support for the robustness of our method under moderate overlap, suggesting that the resulting saliency scores and causal attributions remain stable in such settings.
>
> However, we agree that this empirical evidence does not constitute a formal theoretical guarantee. We consider a more rigorous investigation into the causal implications of overlapping shapelets—particularly with respect to assumptions like SUTVA and ignorability—an important direction for future work.
>
>
> | Model         | MCC-H (overlap)|              |              |
> |-|-|-|-|
> |               | AUPRC          | AUP          | AUR          |
> | ShapeX        | 0.8083±0.0014  | 0.6304±0.0031| 0.7531±0.0027|
>
> ---
>
> > **Q6: Have Cardiologists or Other Domain Experts Assessed the ECG Explanations? Even a Small Pilot Would Strengthen the Practical Claims**
>
> A6: The ECG dataset we use—**PTB-XL**—is annotated by clinical domain experts, and we believe that ShapeX’s strong performance on this dataset indirectly supports the alignment between our method’s explanations and cardiologist-level diagnostic knowledge.
>
> That said, we fully agree that direct expert evaluation would provide more concrete evidence of practical fidelity. Moving forward, we plan to involve **cardiologists** in reviewing and validating the saliency outputs produced by ShapeX. This human-in-the-loop process will enable us to more formally assess the clinical plausibility and trustworthiness of our explanations.
>
> ---
>
> > **Q7: Request for a Summary of Related Works**
>
> A7: Thank you for the helpful suggestion. In the revised manuscript, we will expand the **Related Work** section to provide a more systematic summary of recent time series attribution and explanation methods.

---

### Official Review · Reviewer_DZ2U · 2025-07-01

**Clarity:** 2
**Significance:** 2
**Originality:** 3
**Rating:** 4
**Confidence:** 3

**Summary:**

The paper focuses on explaining a time series predictive model. This is based only on shapelet (time series segments)-driven methods, which makes sense since the focus is on explainable models. Given a set of initial shapelets: For each time series X, the method compares each shapelet S_n with all segments in the time series X and computes alignment scores (descriptors), identifying the most similar segment from X (most strongly aligned) (detectors). The set of shapelets is learnable (although I have questions about this) and optimized to enforce diversity among shapelets and their predictive power. After learning shapelets that can discriminate between classes, a post-hoc analysis is performed to explain the model. This is done by computing Shapley-value attribution. The saliency scores are computed by finding the most strongly aligned segments between all learned shapelets and the time series from the test data. A subset of these segments (g') is perturbed using a mask as in equation 14, and the scores are computed to measure how each segment influences the predictive power of the classifier (equation 12).

**Questions:**

Method:
- not sure why the authors in line 119 define the time series as multivariate (X_t \in R^D) although the method works only on univariate time series.
- how did you obtain the initial set of shapelets? Line 145
- not quite sure how shapelets are learned. Each shapelet is divided into patches and each patch goes through a linear transformation along with multi-head attention across all patches. The question is whether the outcome of the encoder is a representation for each patch or a representation for the entire shapelet? In each case, how does the original set of shapelets get updated? I think what gets updated is how to transform the initial set of shapelets into a useful learnable representation. This needs to be clarified in the manuscript. Additionally, does this depend on the quality of the initial set of shapelets? Does it hinder the explainability of the model? Because initially I thought the explanation would be based on the learned shapelets, but it seems that the initial shapelets are random and it might not be easy to explain the learned transformation of shapelets.
- in equations 7 and 8, what if these time points t are not continuous? For example, for a given shapelet S_n, what if the high activation regions are say 5, 16, 52? These three time points are not contiguous.
- bit confused about the suffix 'i' in equation 23: is it an index for heads or patches?
- it seems from Table 3 that removing the shapelet encoder significantly hinders the performance of the model; AUPRC drops from 72% to 21%. How is the learned transformation of the shapelets used in detecting the most strongly aligned segment? My understanding is that the strongly aligned segments are detected using the initial shapelets**,** not their learned transformation?
- equation 2: Isn't the difference between |g'| and |g| always one? g' is basically g but without shapelet n?
- Equation 11 is not quite clear to me. It needs more explanation.
- what is P(X,M) in line 194?

Experiments:
- all experiments are performed on binary classification problems, although the method was proposed to be general for multi-class classification.
- from Figure 4, it seems that dyna is more robust to highly perturbed segments than ShapeX. Is there any explanation?
- Fails to compare against published post-hoc explainers for shapelet-based methods such as
	- Bahri, Omar, Soukaina Filali Boubrahimi, and Shah Muhammad Hamdi. "Shapelet-based counterfactual explanations for multivariate time series." arXiv preprint arXiv:2208.10462 (2022).
	- Mochaourab, Rami, et al. "Post hoc explainability for time series classification: Toward a signal processing perspective." IEEE signal processing magazine 39.4 (2022): 119-129.
	- Huang, Qi, et al. "Shapelet-based model-agnostic counterfactual local explanations for time series classification." arXiv preprint arXiv:2402.01343 (2024).
	- Meng, Fanyu, et al. "Implet: A Post-hoc Subsequence Explainer for Time Series Models." arXiv preprint arXiv:2505.08748 (2025).

I do not think that shapelet-based methods need any post-hoc explanation as shapelets are inherently interpretable. The proposed method fits in the area of attribution more than post-hoc explaination.

**Ethical Concerns:**

["NO or VERY MINOR ethics concerns only"]

**Final Justification:**

Authors replied to my concerns.

**Quality:**

2

**Strengths And Weaknesses:**

Strength:
- Explainable AI models are of utmost importance, especially for time series analysis
- Based on well known shapelet-driven time series framework

Weakness:
- If I'm not mistaken the main contribution of the paper is basically in equation 12, which is minor and incremental.
- Proving shapelet-value is equivalent to model-level CATE is straight-forward.
- Fail to compare against other post-hoc explainer that are based on shapelet-driven methods.

---

> ### Author Rebuttal · Authors · 2025-07-31
>
> We sincerely thank you for your careful reading of our work and the constructive comments provided. Your feedback has significantly helped us improve the clarity and positioning of our manuscript.
>
> > **Q1: Ambiguity in Defining Time Series as Multivariate While Focusing on Univariate Case**
>
> A1: We agree that this may appear confusing. Our intention was to present a general formulation in the methodology section to maintain future extensibility to multivariate time series. However, as explicitly stated in line 125, this paper focuses solely on the univariate setting ($D = 1$). We will revise the manuscript to make this restriction clearer earlier in the main text to avoid misunderstanding. In addition, we **have included new experiments during the rebuttal phase to demonstrate that ShapeX can be easily extended to multivariate time series**. For the specific experimental setup and results, please refer to our response to Reviewer NMJu’s Q2.
>
> ---
>
> > **Q2: Clarification on How the Initial Shapelet Set Is Obtained**
>
> A2: The initial shapelets are **randomly initialized**. The detailed initialization process is described in Appendix C. We acknowledge that this is a critical design detail and should have been clearly stated in the main text. We sincerely apologize for this oversight and will make sure to highlight it explicitly in the revised manuscript.
>
> ---
>
> > **Q3: How Are Shapelets Learned and Are They Sensitive to Initialization Quality?**
>
> A3: The shapelets are learned via a self-attention-based encoder, following a structure similar to PatchTST. Each shapelet is divided into patches, passed through linear transformations, and processed by multi-head attention to produce a unified representation for the entire shapelet—not for individual patches.
>
> This design is briefly described in Appendix C, but we acknowledge that it deserves clearer exposition in the main text and will revise accordingly.
>
> To examine whether the initial shapelet quality impacts the results, we conducted a robustness study using three different random seeds for initialization. As shown below, the **explanation quality exhibit negligible variation, suggesting that the learned shapelets are stable and insensitive to initialization**.
>
>
>
> | Seed       | MCC-E         |             |             | ECG       |             |             |
> |-|-|-|-|-|-|-|
> |            | AUPRC          | AUP         | AUR         |AUPRC           | AUP         | AUR         |
> | seed=44| 0.6407±0.0036  |0.5614±0.0076  |  0.3679±0.0050 |   0.7228±0.0028    |   0.8395±0.0030   | 0.6961±0.0032 |
> | seed=22| 0.6443±0.0012  |0.5623±0.0004  |  0.3716±0.0069 |   0.7205±0.0012    |   0.8320±0.0059   | 0.6924±0.0009 |
> | seed=5 | 0.6402±0.0030  |0.5604±0.0054  |  0.3681±0.0043 |   0.7340±0.0031    |   0.8334±0.0049   | 0.6956±0.0045 |
>
> ---
>
> > **Q4: Concern About Discontinuity in Activated Time Steps in Equations 7 and 8**
>
> A4: This is an excellent observation. In practice, the high-activation regions of a shapelet $S_n$ are often non-contiguous. Therefore, $X(S_n)$ does not necessarily refer to a continuous segment—it may consist of disjoint time indices that align strongly with $S_n$.
>
> This does not affect our downstream steps, such as perturbation and Shapley value calculation. However, we agree that the current form of Equations (7) and (8) may imply continuity and thus be misleading. We will revise the manuscript to explicitly clarify that non-contiguous activations are valid and expected.
>
> ---
>
> > **Q5: Clarification on the Meaning of Index 'i' in Equation 23**
>
> A5: In Equation 23, the index $i$ refers to different attention heads. We will clarify this explicitly in the revised manuscript to avoid potential ambiguity.
>
> ---
>
> > **Q6: Are Strongly Aligned Segments Based on Initial or Learned Shapelets?**
>
> A6: This is a very reasonable concern. The misunderstanding may stem from our lack of clarity in explaining that the shapelets are initially random and are then learned through the encoder. If we used the initial untrained shapelets, they would have no meaningful structure and could not align with any relevant segments.
>
> The encoder is crucial—it transforms randomly initialized sequences into meaningful shapelets that can align with semantically important regions in the input. Thus, the strongly aligned segments are based on the learned shapelets, not the initial ones.
>
> ---
>
> > **Q7: Does the Difference Between $|G'|$ and $|G|$ Always One?**
>
> A7: We believe you are referring to Equation 12. In this context, $G'$ is defined as an arbitrary subset of $G \setminus \{n\}$—that is, any subset of $G$ excluding the element $n$—and is not necessarily of size $|G| - 1$. This definition is indicated below the summation symbol in the equation. We acknowledge that this notation may be unintuitive at first glance, and we will revise the manuscript to make the definition of $G'$ more explicit and accessible.
>
> ---
>
> > **Q8: Lack of Clarity in Equation 11**
>
> A8: The goal of Equation 11 is to apply linear interpolation between $X_{\text{start}}$ and $X_{\text{end}}$ so that the pertubation is smooth. This design helps prevent introducing abrupt, unrealistic changes during perturbation. We will add a more detailed explanation of this operation and its intent in the revised manuscript.
>
> ---
>
> > **Q9: What Is $P(X, M)$ in Line 194?**
>
> A9: $P(X, M)$ refers to the perturbation function defined in Equation 14 and elaborated in the appendix. We will add a clear reference in the main text to avoid confusion.
>
> ---
>
> > **Q10: All Experiments Use Binary Classification**
>
> A10: Thank you for pointing this out. In fact, 85 of the UCR Archive datasets used in our experiments are multi-class classification tasks. **This was not explicitly annotated in Table 4, and we will correct this oversight in the revised manuscript to avoid misinterpretation.**
>
> ---
>
> > **Q11: From Figure 4, it seems that dyna is more robust to highly perturbed segments than SHAPEX. Is there any explanation?**
>
> A11: You have a very keen observation. Figure 4 indeed shows results from two randomly selected datasets in the UCR Archive. However, as shown in **Figure 6 and Figure 8–11** in the appendix, the complete evaluation across all 112 datasets reveals that it is difficult to conclude that dyna is consistently more robust.
>
> That said, in certain datasets, dyna does show smoother performance degradation curves than ShapeX. We suspect this is because dyna computes saliency scores at the single timestep level, which leads to more scattered and discrete saliency regions. When perturbed, these regions result in more gradual degradation. However, as shown in Figure 6, although Dyna’s curve appears smoother, its performance at nearly every perturbation ratio is worse than ShapeX, and it degrades more sharply in early perturbation stages.
>
> ---
>
> > **Q12: Fails to Compare Against Published Shapelet-Based Methods**
>
> A12: Thank you very much for providing these valuable references. We **have included Implet [Meng et al., 2025] as a baseline in our comparative experiments—results are reported below**. Unfortunately, we were unable to include the other cited methods due to the lack of publicly available code. If these works release their implementations in the future, we would be glad to incorporate them into future evaluations.
>
> From the experimental results, ShapeX consistently outperforms Implet.
>
> ---
>
> | Model         | MCC-E          |              |              | MCC-H          |              |              |
> |-|--|-|-|--|-|-|
> |               | AUPRC          | AUP          | AUR          | AUPRC          | AUP          | AUR          |
> | ShapeX        | 0.6407±0.0036  | 0.5614±0.0076| 0.3679±0.0050| 0.8110±0.0003  | 0.6838±0.0054| 0.7431±0.0055|
> | Implet        | 0.3701±0.0025  | 0.5184±0.0020| 0.4901±0.0029| 0.5103±0.0079  | 0.4801±0.0038| 0.5501±0.0028|
>
> | Model         | MTC-E          |              |              | MTC-H          |              |              |
> |-|--|-|-|--|-|-|
> |               | AUPRC          | AUP          | AUR          | AUPRC          | AUP          | AUR          |
> | ShapeX        | 0.6100±0.0048  | 0.3962±0.0067| 0.5472±0.0082| 0.6792±0.0014  | 0.4255±0.0024| 0.9019±0.0041|
> | Implet        | 0.1740±0.0023  | 0.1802±0.0009| 0.5082±0.0011| 0.3922±0.0062  | 0.5024±0.0012| 0.4502±0.0035|
>
> | Model         | ECG            |              |              |
> |-|--|-|-|
> |               | AUPRC          | AUP          | AUR          |
> | ShapeX        | 0.7228±0.0028  | 0.8395±0.0030| 0.6961±0.0032|
> | Implet        | 0.5385±0.0035  | 0.5092±0.0024| 0.4845±0.0042|
>
> ---
>
> > **Q13: Whether Post-Hoc Explanations Are Necessary for Shapelet-Based Methods, and Whether the Proposed Method Is Better Categorized as Attribution**
>
> A13: We sincerely appreciate this thoughtful and insightful comment. Indeed, our method relies on shapelets, which are inherently interpretable. However, in ShapeX, **shapelets serve purely as an intermediate representation to facilitate the computation of saliency scores**, and are not used for classification or as an interpretability tool themselves.
>
> The goal of ShapeX is to explain the behavior of a black-box model, which aligns with the spirit of post-hoc explanation. In fact, our use of the term “post-hoc” follows the convention established in TimeX++ under the phrase “Post-hoc Instance-level Time Series Explanation.” However, we agree with the reviewer that the overall design of ShapeX—especially its emphasis on saliency estimation and perturbation analysis—fits naturally within the attribution literature.
>
> Thus, **ShapeX can reasonably be viewed as both a post-hoc explanation tool and an attribution method**. To better clarify its position to readers, we **will categorize ShapeX under the umbrella of attribution methods in the revised manuscript**, particularly in the introduction and methodology sections.

---

> > ### Author Response · Authors · 2025-08-01
> > **Clarifying the Core Contribution of ShapeX**
> >
> > We offer a brief clarification here to address a possible misinterpretation of our **main contribution** noted in the Weakness section.
> >
> > The core contribution of **ShapeX** lies in the **first** introduction of a shapelet-driven, **segment-level feature attribution** explanation paradigm for time series models.
> > We consider this a **significant breakthrough** in time series explanability, as it overcomes the limitations of existing methods that almost exclusively rely on **timestep-level explanations**, and offers an attribution mechanism with **stronger semantic consistency and causal interpretability**.
> >
> > Specifically, our approach goes **far beyond a mere reformulation of Shapley Value computation**.
> > Instead, we propose a pioneering
> >  explanatory framework that unifies **shapelet learning**, **structural segmentation**, and **causal attribution**.
> > The key contributions include:
> >
> > 1. **A novel explanation paradigm**
> >    We are the **first to bring shapelets**—traditionally used for inherently interpretable models—into the **attribution explanation** setting.
> >    By leveraging shapelet-driven structural segmentation, we **organically integrate shapelet representations with Shapley-based attribution**.
> >    This not only **expands the research boundaries of shapelets** but also **opens up a new avenue** for time series attribution.
> >
> > 2. **The most extensive experimental evaluation in this area**
> >    In addition to widely used datasets in this domain, we conduct, **for the first time**, occlusion experiments on over **100 UCR datasets**.
> >    We thoroughly validate the superiority of **ShapeX** in terms of **accuracy**, **robustness**, and **causal fidelity**, thereby establishing **a new benchmark** and **laying a solid foundation for future research** in this field.

---

> > > ### Author Response · Authors · 2025-08-06
> > >
> > > We would like to kindly check in to see if the reviewer has any remaining questions or feedback regarding our response.
> > > We are happy to provide any further clarification if needed.
> > >
> > > Thank you again for your time and consideration.

---

> > ### Comment · Reviewer_DZ2U · 2025-08-07
> > **Calrification recevied**
> >
> > Thanks for answering my points. Now I have a good understanding of the method. I will raise my assessment.

---

### Official Review · Reviewer_vy4B · 2025-07-01

**Clarity:** 2
**Significance:** 2
**Originality:** 2
**Rating:** 4
**Confidence:** 5

**Summary:**

This paper presents ShapeX, a post-hoc time series explanation framework which segments time series into meaningful subsequences denoted as shapelets and uses shapley values to determine their saliency. The framework uses a describe and detect method which learns shapelets meaningful for classification. The framework is compared against benchmarks on 4 synthetic and 100+ real datasets from the UCR archive.

**Questions:**

1. How to guarantee the validity or “truthfulness” of the shapelets generated by the ShapeX framework compared to the implicit shapelets used by the classification model? What if these are not the same – i.e., the black box classification model uses a different set of shapelets than those identified by the ShapeX framework?

2. How to guarantee that the shapelets actually represent subsequences from the real raw input time series? It looks like the generated shapelets may not even correspond to the real underlying time series (as seen in Figure 7 where the 4 shapelets do not directly match subsequences of the input time series.)

3. The experimental results seem to use an apples to oranges comparison where the saliency at each timestep was derived from multiple timesteps in ShapeX and prior methods only have information from a single timestep. Can the authors justify this comparison?

**Ethical Concerns:**

["NO or VERY MINOR ethics concerns only"]

**Final Justification:**

Given the authors detailed rebuttal, and taking into account the other reviews and comments, my main concerns surrounding issues with selecting shapelets in the methods, validity of the experiments, and structural organization of the paper have been alleviated. Therefore I raise my score to a 4.

**Limitations:**

Although Section 6 Conclusion briefly acknowledges a limitation, it appears to be a rather superficial and self-evident one (use for univariate time series only). Given the complexity of the proposed framework, a more thorough and insightful analysis of the framework's limitations would strengthen the paper.

**Quality:**

3

**Strengths And Weaknesses:**

Overall, this paper highlights an important gap in current research for explainability methods in time series classification but has fundamental issues in the methodology and results.

The premise of the paper is interesting and important, namely that the impact of time series subsequences i.e., shapelets, are more significant for explaining classification models than individual timesteps. All prior work focuses on individual timestep explanations, with nothing focusing on providing explanations for trace subsequences or shapelets. This is a very important gap in current literature. Therefore, the development of a new framework to address this gap is highly original with the potential for high significance and impact.

However, I believe there are fundamental issues with the methodology and experimental results. I have concerns about the shapelet describe and detect framework in which shapelets are self-identified and generated *independently* from the black-box classification model. This appears to be an issue because there could be a mismatch between the shapelets identified by the ShapeX framework, and those actually used by the classification model. For example, the transformation of time series used in classification models (e.g., the convolutional kernels used in MultiRocket) may be different from the transformation used in the descriptor component of ShapeX (e.g., the generated activation maps). There is no guarantee that the generated shapelets actually correspond to the underlying subsequences used in the classification model; it seems these two should be directly tied to each other. (See Question 1). I am further concerned because it appears that the generated shapelets may not even correspond to the real underlying time series for example as seen in Figure 7 where the 4 shapelets do not directly match subsequences of the input time series. (See Question 2).

The experimental results do not appear technically sound. It seems there is an unfair comparison between ShapeX, that computes explainability/saliency for a *set* of timesteps, and other prior methods which compute explainability/saliency for a *single* timestep. In ShapeX according to Equation 19, to obtain saliency scores for each individual time step, the segment level shapley value is uniformly disturbed across the time steps within each segment. This means the learned explainability value at each timestep is derived from information it learned from *more* than one timestep, when compared to prior methods that only have knowledge about a *single* timestep (e.g., like in Dynamask which builds a perturbation for each feature at each individual timestep). This seems to be an apples to oranges comparison, where ShapeX has an unfair advantage (i.e., the saliency was given extra context from timesteps before or after the current one) compared to prior methods that only have knowledge about a *single* timestep. A fairer comparison might also aggregate the values from the same trace subsequences in the benchmarks so they can be more directly compared. (See Question 3). Moreover, while it's commendable that the authors evaluated their approach on 100+ datasets from the UCR archive, I have concerns about the validity of reporting results averaged across all datasets. This approach overlooks the unique characteristics of each dataset and may obscure important nuances, especially when large averages mask dataset-specific performance differences. This is further supported by the very high error in the presented results (for example Figure 6(a) has very large error bars across all methods including ShapeX).

The paper also has issues with clarity. The paper is extremely dense and hard to parse as a result of many important technical descriptions related to the methodology and the experiments being in the appendix (including description of encoder in Appendix C, saliency score calculation in equation 19 in Appendix A.2, metric definitions in Appendix E, testing on multiple black-box models in Appendix F). The large number of nonstandard abbreviations used (e.g., SV, SDD, SDSL, PHTSE) contribute to the paper being hard to parse as well. It is unclear to me the point of Section 4 – it seems this would be better removed or placed in the appendix and instead this space used to further describe important technical methodology and experimental choices that are currently in the appendix. Some figures and tables are confusing: Table 1 and 2 are confusing and hard to read with all of the blue shading; it is unclear what the expected conclusion we should draw from Figure 6 (b) is.

---

> ### Author Rebuttal · Authors · 2025-07-31
>
> Thank you for these thoughtful suggestions. We believe **many of the concerns raised stem from our suboptimal separation of content between the main text and appendix**. We are confident that **through better organization and clearer presentation, we can resolve the majority of these concerns** and significantly improve the accessibility and clarity of the paper.
>
> > **Q1: How to Ensure the Validity of ShapeX Explanations When Its Learned Shapelets May Differ from the Implicit Representations Used by the Black-Box Model**
>
> A1: We sincerely appreciate your insightful question, which touches on the core of ShapeX's design philosophy. We acknowledge that this question appears very reasonable and likely stems from a **misunderstanding caused by not defining the problem setting in the clearest and most rigorous terms**.
>
> First, from a problem formulation perspective, the task we address is well-defined: given a time series input and a trained black-box classifier, we aim to assign importance scores to input time series, where a higher score indicates greater influence on the model's decision. In this paradigm, **the shapelets identified by ShapeX are not assumed to coincide with the internal representations of the black-box model.** Instead, they are treated as **intermediate constructs** to facilitate effective segment-level attribution.
>
> To be transparent, ShapeX can be interpreted as hypothesizing what model-relevant shapelets might be. If these learned shapelets align well with the truly discriminative patterns used by the model, the resulting saliency score is accurate. **However, even if the guessed shapelets do not precisely match the model’s internal logic, the final saliency score can still remain robust.** This robustness stems from the flexible structure of ShapeX: **the learned shapelets guide the preliminary segmentation**, while the **final saliency scores are computed from perturbation-based outcomes**, independently of whether the segmentation is perfectly aligned with the model's actual reasoning.
>
>
> To validate this claim, we designed a controlled experiment using ShapeFormer as the black-box model to be explained. We then created a variant of ShapeX, denoted **ShapeX-S**, in which we replaced the learned shapelets in ShapeX with the exact shapelets used internally by ShapeFormer for segmentation.
> In this setup, ShapeX-S uses the **exact same shapelets as the black-box model**, ensuring full alignment between the segmentation patterns and the model's internal logic.
> The experimental results show that **ShapeX-S often performs worse than the original ShapeX**, which uses independently learned shapelets. This finding suggests that our method **does not require perfect alignment with the black-box model’s internal representations** to produce accurate and faithful explanations.
> We believe this supports our core hypothesis: ShapeX explanations rely more on the perturbation-based attribution mechanism than on precise imitation of the model’s internal structure.
>
> | Model       | MCC-E         |             |             | ECG       |             |             |
> |-|-|-|-|-|-|-|
> |            | AUPRC          | AUP         | AUR         |AUPRC           | AUP         | AUR         |
> | ShapeX  | 0.6407±0.0031  |0.5614±0.0076  |  0.3679±0.0050 |   0.7228±0.0028    |   0.8395±0.0030   | 0.6961±0.0032 |
> | ShapeX-S| 0.5782±0.0056  |0.6027±0.0007  |  0.2513±0.0009 |   0.6525±0.0044    |   0.7435±0.0075   | 0.5730±0.0021 |
>
>
> ---
>
> > **Q2: How to Ensure That the Learned Shapelets Correspond to Real Subsections of the Raw Input Time Series**
>
> A2: This is a very natural and reasonable concern, and we thank the reviewer for the opportunity to clarify this point. In early shapelet-based literature (e.g., [40]), shapelets were indeed defined as **actual subsequences** extracted directly from the raw input time series. However, more recent advances—including our work—adopt a **more flexible and generalized definition**, in which **shapelets are learned representations obtained through optimization or deep neural networks** [41,42]. We note that the reference numbers are consistent with those in the main submission.
>
> Under this modern formulation, the shapelets produced by ShapeX are **not intended to exactly replicate raw input subsequences**. Instead, they are **abstract, representative patterns** that best capture discriminative structures across multiple input samples. These learned shapelets act as **pattern templates** rather than literal time series fragments.
>
> We acknowledge that **the confusion may stem from the fact that we placed this foundational clarification in Appendix A.4**, which may have made it less accessible to readers. This was an oversight on our part. In the revised manuscript, we will move the definition and conceptual framing of shapelets to the main methodology section to prevent similar misunderstandings.
>
> ---
>
> > **Q3: Justification for Comparing ShapeX’s Segment-Level Saliency with Prior Per-Timestep Methods**
>
> A3: This is a very insightful and important observation. It raises a fundamental question that is rarely addressed in the time series explainability literature: **What exactly is the task formulation, and what types of temporal dependencies are permissible in computing saliency?**
>
> Our problem setting follows the formulation used in prior works such as DynaMask, TimeX, and TimeX++, as described in Section 3.1. This task formulation is deliberately broad—it specifies the overall input-output behavior of the explainer, but **does not restrict whether the saliency score for a specific timestep must be based solely on that timestep or can incorporate surrounding context.**
>
> In fact, **several baselines already utilize temporal dependencies.** For example:
> - **TimeX** employs a Transformer-based encoder-decoder that takes the entire sequence as input to generate binary attribution masks for each timestep. It also introduces smoothness constraints to enforce local continuity in saliency scores.
> - **TimeX++** similarly uses a Transformer encoder to model cross-timestep dependencies, with its Smoothness Regularization explicitly encouraging temporally coherent attributions.
>
> Therefore, we believe that ShapeX’s segment-level attribution mechanism remains consistent with the problem setting adopted in existing methods. If anything, your question highlights a broader challenge in the field—the **lack of standardized assumptions** about how attribution across time should be formulated.
> In response, we **will add a dedicated paragraph in the Related Work section of our revised manuscript to clarify how different methods interpret and implement this task formulation.** We greatly appreciate you bringing this issue to our attention.
>
> ---
>
> > **Q4: Request for a More Thorough and Insightful Analysis of ShapeX’s Limitations Beyond the Univariate Setting**
>
> A4: We agree that the current manuscript’s treatment of limitations—focused only on the univariate setting—is relatively superficial. A more thorough discussion will certainly strengthen the clarity and scope of the work.
>
> In addition to the univariate constraint, ShapeX has the following key limitations:
>
> - **Predefined hyperparameters**: The framework requires several user-defined parameters, including the threshold $\Omega$, the number of shapelets, and their lengths. These hyperparameters may affect performance and require tuning for different datasets.
>
> - **Training requirement**: Like TimeX and TimeX++, ShapeX involves a separate training stage to learn shapelets and optimize the SDD module. This reduces flexibility compared to gradient-based methods like Integrated Gradients, which compute saliency scores directly without training.
>
>
> We will explicitly incorporate these points into Section 6 of the revised manuscript to provide a more balanced and transparent evaluation of our framework.
>
> ---
>
> > **Q5: Concern About Averaging Results Across UCR Datasets and Large Variance Obscuring Dataset-Specific Differences**
>
> A5: Thank you for highlighting this important concern. We agree that averaging performance across 100+ UCR datasets may obscure meaningful dataset-specific behaviors.
>
> We **have provided** detailed results for **each individual UCR dataset in Figures 8–11**. These plots show the per-dataset perturbation curves, allowing readers to examine how ShapeX and other methods perform under varying datasets on a case-by-case basis.
>
> We will also revise the caption of Figure 6 in the main text to explicitly direct readers to the appendix, ensuring that those interested in dataset-level granularity can easily locate and interpret the relevant figures. Thank you again for pointing this out.
>
> ---
>
> > **Q6: Clarification Requested Regarding Presentation and Organization of Key Content**
>
> A6: We appreciate the reviewer’s feedback on presentation quality and acknowledge that certain important technical components were placed in the appendix, which may have inadvertently led readers to overlook critical context.
>
> In the revised manuscript, we will reorganize the structure to surface essential elements—such as the definition of shapelets and the introduction of the shapelet encoder—into the main body. Less central implementation details will remain in the appendix for completeness.
>
> Additionally, we will refine terminology usage by avoiding unnecessary abbreviations and using full spellings for commonly recognized terms (e.g., writing out “Shapley Value” in full). We will also improve the clarity of figure captions. For example:
> - In **Tables 1 and 2**, we will clarify that the color shading reflects the relative magnitude within each column;
> - In **Figure 6(b)**, we will state explicitly that each point corresponds to the best-performing explainer for the respective dataset at a given perturbation ratio.

---

> > ### Comment · Reviewer_vy4B · 2025-08-04
> >
> > Thank you for your responses to my questions. I've reviewed your clarifications, but I still have significant concerns regarding points 5 and 6.
> >
> > Averaging Results Across UCR Datasets: I appreciate you pointing out the per-dataset figures (8-11). However, these reinforce my concern. While there are instances where ShapeX outperforms other models, there are also many cases where it underperforms, and even some where it performs the worst. For example, in Figure 10, it struggles with MIDDLEPHALANXTW and POWERCONS, and in Figure 11, it is the worst for TWOLEADECG and SMOOTHSUBSPACE. The inconsistent performance and wide error bars across such a diverse set of datasets makes the averaged results misleading. I do not believe this method of reporting is a valid representation of the model's overall performance.
> >
> > Presentation and Organization of Key Content: I agree with your acknowledgment that the paper's organization led to significant confusion. It seems that the necessary revisions to properly integrate key content from the appendix into the main text would require a major restructuring of the paper. It seems this would be a substantial undertaking that I believe is beyond the scope of a typical revision, and would require a new submission for a full, cohesive review.

---

> > > ### Author Response · Authors · 2025-08-05
> > >
> > > We sincerely thank you for your engagement and valuable feedback. We greatly appreciate your follow-up comments on **Points 5 and 6**, and we address them below in turn.
> > >
> > > ---
> > >
> > > ### Response to Point 5: On Averaging Results Across UCR Datasets
> > >
> > > We would like to clarify that our evaluation of explanation methods in this paper consists of two complementary perspectives:
> > > - **Direct evaluation** using datasets that provide ground-truth saliency labels (e.g., MCC-E, MTC-E, MCC-H, MTC-H, and ECG), and
> > > - **Indirect evaluation** through occlusion perturbation experiments on the UCR archive.
> > >
> > > Therefore, our primary conclusions are drawn from the **direct evaluations on datasets with ground-truth saliency**, while the occlusion results over UCR are intended to **complement these findings** by assessing the generalizability and robustness of the explanation method in broader settings.
> > >
> > > Regarding your concern that ShapeX underperforms on some individual datasets, we would like to provide the following summary statistics:
> > > Across **112 UCR datasets**, with **14 occlusion ratios per dataset**, we obtain **1,680 experimental configurations** in total. The number of configurations in which each method achieves the best performance is shown below:
> > >
> > > | Model    | Wins | Ratio   |
> > > |----------|------|---------|
> > > | ShapeX   | 731  | 43.5%   |
> > > | Dyna     | 379  | 22.6%   |
> > > | TIMEX    | 228  | 13.6%   |
> > > | Random   | 186  | 11.1%   |
> > > | TIMEX++  | 156  | 9.3%    |
> > >
> > > These results demonstrate that **no method consistently dominates across all settings**, which is also a known phenomenon in UCR classification tasks.
> > > Nonetheless, **ShapeX outperforms all other methods in 43.5% of cases—nearly twice that of the second-best model**. We believe this provides strong evidence of the robustness and wide applicability of our method across diverse tasks.
> > >
> > > ---
> > >
> > > ### Response to Point 6: On Presentation and Organization of Key Content
> > >
> > > We are grateful for your suggestions regarding the presentation and organization of the manuscript. Based on your feedback, we have made the following structural improvements in our working draft:
> > >
> > > 1. We have moved the original Section 4 (ShapeX as Approximate Causal Attribution) to the appendix, allowing the main methodology section to focus exclusively on the ShapeX framework, thereby improving clarity and readability.
> > > 2. We have integrated the definition of shapelets into Section 3.1, and consolidated the encoder details into a dedicated subsection titled "Shapelet Encoder and Training Losses."
> > > 3. We have moved the formulation of the saliency score (previously Eq. 19) into the main text, placing it directly after Eq. 12.
> > > 4. We have relocated the description of evaluation metrics from Appendix E.2 to the beginning of the *Experiments* section.
> > >
> > > **These changes have already been implemented in our internal draft, and we did not encounter any significant difficulties during restructuring.** However, due to submission constraints, we are currently unable to share the revised version.
> > >
> > > That said, we are **confident that the camera-ready version will present a significantly improved manuscript with clear structure and strong readability.** Given that reviewers **NMJu** and **zzMH** have both explicitly found the current version to be well organized, we believe the revised structure will be well within the scope of a camera-ready revision.

---

> > > > ### Comment · Reviewer_vy4B · 2025-08-05
> > > >
> > > > Thanks for your clarification, this was very helpful and alleviated my concerns surrounding points 5 and 6. Taking into account your additional context (particularly the model comparison table and delineation of structural changes to the paper) and the other reviewers comments, I will raise my score.

---

> > > > > ### Author Response · Authors · 2025-08-05
> > > > >
> > > > > Thank you very much for your thoughtful follow-up and for reconsidering your evaluation.
> > > > >
> > > > > We truly appreciate your feedback and are happy to address any further questions or concerns you may have.

---

### Decision · Program_Chairs · 2025-09-17

**Decision:**

Accept (poster)

**Comment:**

**Summary**
Existing post-hoc time series explanation methods primarily focus on timestep-level feature attribution, and overlook the fundamental prior that classification outcomes are predominantly driven by key shapelets. This paper presents ShapeX, which first segments time series into shapelet-driven segments and then employs Shapley values to assess their saliency. On both synthetic and real-world datasets, ShapeX produces explanations revealing causal relationships instead of just correlations.

**Strengths**
- The premise of this paper (that the impact of shapelets are more significant for explaining classification models than individual timesteps) is novel, interesting and important.
- The authors evaluated their approach on 100+ datasets from UCR archive.
- The joint learning of shapelets with matching + diversity losses improves fidelity, and there are thorough ablations that isolate each component's impact.

**Weaknesses**
- In the shaplet describe and detect (SDD) framework, shapelets are self-identified and generated independently from the black-box classification model.
- The comparison of ShapeX to traditional step-level explanation might have unfair advantage to ShapeX, considering that the attribution scores of ShapeX aggregates over multiple timesteps. However, the authors point out that the attribution scores of multiple step-level explanations also utilize cross-timestep dependencies.
- Human-evaluated interpretability usefulness would be good-to-have.
- The computational cost of Shapley sampling in wall-clock terms would be good-to-have.
- More discussions of related works, and improvements on the writing would improve the presentation of this paper.